# Metamorphic gene regulation programs in *Xenopus tropicalis* tadpole brain

**Samhitha Raj[1], Christopher J. Sifuentes[1¤a], Yasuhiro Kyono[2¤b], Robert J. Denver[1,2]***

**1** Department of Molecular, Cellular and Developmental Biology, University of Michigan, Ann Arbor, Michigan, United States of America, **2** Neuroscience Graduate Program, University of Michigan, Ann Arbor, Michigan, United States of America

¤a Current address: Chan Zuckerberg Initiative, Redwood City, CA, United States of America
¤b Current address: Department of Pharmacy Practice, College of Pharmacy, University of Illinois at Chicago, Chicago, IL, United States of America
* rdenver@umich.edu

**Data Availability Statement:** The RNA-seq datasets used for the current analyses were previously published and the data deposited in GEO (developmental: GSE139267; 3,5,3'-triiodothyronine (T3)-induced: GSE130816). The

## Abstract

Amphibian metamorphosis is controlled by thyroid hormone (TH), which binds TH receptors (TRs) to regulate gene expression programs that underlie morphogenesis. Gene expression screens using tissues from premetamorphic tadpoles treated with TH identified some TH target genes, but few studies have analyzed genome-wide changes in gene regulation during spontaneous metamorphosis. We analyzed RNA sequencing data at four developmental stages from the beginning to the end of spontaneous metamorphosis, conducted on the neuroendocrine centers of *Xenopus tropicalis* tadpole brain. We also conducted chromatin immunoprecipitation sequencing (ChIP-seq) for TRs, and we compared gene expression changes during metamorphosis with those induced by exogenous TH. The mRNA levels of 26% of protein coding genes changed during metamorphosis; about half were upregulated and half downregulated. Twenty four percent of genes whose mRNA levels changed during metamorphosis had TR ChIP-seq peaks. Genes involved with neural cell differentiation, cell physiology, synaptogenesis and cell-cell signaling were upregulated, while genes involved with cell cycle, protein synthesis, and neural stem/progenitor cell homeostasis were downregulated. There is a shift from building neural structures early in the metamorphic process, to the differentiation and maturation of neural cells and neural signaling pathways characteristic of the adult frog brain. Only half of the genes modulated by treatment of premetamorphic tadpoles with TH for 16 h changed expression during metamorphosis; these represented 33% of the genes whose mRNA levels changed during metamorphosis. Taken together, our results provide a foundation for understanding the molecular basis for metamorphosis of tadpole brain, and they highlight potential caveats for interpreting gene regulation changes in premetamorphic tadpoles induced by exogenous TH.

TR ChIP-seq dataset has been deposited in GEO (GSE214697).

**Funding:** This research was supported by National Science Foundation grant IOS 1456115 to RJD. https://www.nsf.gov/ The funders had no role in study design, data collection and analysis, decision to publish, or preparation of the manuscript.

**Competing interests:** The authors have declared that no competing interests exist.

## Introduction

Most amphibian species have complex life cycles with two discrete life history stages separated by a metamorphosis. Amphibian metamorphosis is controlled by thyroid hormone (TH), which binds to ligand-activated transcription factors (TH receptors—TRs) to regulate gene transcription [1]. All vertebrates studied to date have two genes that code for TRs, designated alpha and beta. The TRs bind to DNA as heterodimers with retinoid X receptor at TH response elements (TREs), which are comprised of two, six nucleotide stretches called half sites; the most common type of TRE has two half sites in direct repeat with a four base spacer (DR+4) [2]. The TH-TR complexes induce biochemical, physiological and morphological changes during metamorphosis by controlling transcription of genes encoding proteins that underlie tissue transformations [3]. Thyroid hormone biosynthesis and release is regulated by neuroendocrine centers in tadpole brain whose maturation depends on TH [4,5]. Neurosecretory neurons located in the anterior preoptic area (POa) and hypothalamus produce neurohormones that are released into the pituitary portal system where they travel to the adenohypophysis to regulate secretion of thyroid stimulating hormone, which then acts on thyroid follicle cells to induce TH biosynthesis and release [4].

To understand the molecular basis for tadpole metamorphosis, investigators have used gene expression screening techniques like subtractive hybridization, differential display, DNA microarray, and recently, RNA sequencing to identify genes regulated by TH in tadpole tissues. These experiments identified genes that encode proteins involved with TH signaling like TRs, monodeiodinases, chromatin modifiers, and transcription factors that mediate TH actions, among others [6–24]. While these studies have illuminated some of the cellular pathways regulated by TH, and have provided sets of genes for further study, most relied on treatment of premetamorphic tadpoles with TH versus analysis of spontaneous metamorphosis. Therefore, we currently lack a comprehensive understanding of the molecular changes that drive the transformation of tadpole tissues during spontaneous metamorphosis.

We recently published an RNA-seq dataset generated from *Xenopus tropicalis* tadpole brain (region of the preoptic area/thalamus/hypothalamus) at four developmental stages encompassing the beginning to the end of spontaneous metamorphosis [25]. In that study we investigated the relationship between DNA methylation and gene regulation changes during metamorphosis, but we did not analyze cellular pathways represented in the RNA-seq dataset. Ta and colleagues [24] recently reported an RNA sequencing experiment conducted on the midbrain of *Xenopus laevis* tadpoles at four developmental stages from premetamorphosis to just before the climax of metamorphosis. To our knowledge, there have been no comprehensive, genome-wide analyses of gene expression changes in any tadpole tissue that spanned the start to the end of spontaneous metamorphosis, nor has there been a comparison of these changes with those induced by exogenous TH. Such analyses are essential to understand the gene regulation programs during spontaneous metamorphosis, and to determine if the gene expression changes induced by exogenous TH in premetamorphic tadpoles reflect what occurs during normal development.

In the current study, we investigated gene regulation changes in neuroendocrine centers of *X. tropicalis* tadpole brain during spontaneous metamorphosis. We analyzed RNA-seq data [25] collected at four stages from the beginning to the completion of metamorphosis. We also conducted chromatin precipitation sequencing (ChIP-seq) for TR on tadpole brain chromatin at metamorphic climax when circulating TH and expression of TRs are maximal. Lastly, we compared gene regulation changes during spontaneous metamorphosis to the set of genes modulated by treatment of premetamorphic tadpoles with 3,5,3'-triiodothyronine (T$_3$) for 16 h [21]. Our findings provide essential data for understanding the molecular basis for

metamorphosis of tadpole brain, and also demonstrate potential caveats for interpreting data from TH-induced gene regulation studies conducted on premetamorphic tadpoles.

## Materials and methods

### Animal care and use

We obtained *X. tropicalis* tadpoles by in-house breeding, reared them in dechlorinated tap water (water temperature 25°C, pH 7) and maintained them on a 13L:11D photoperiod. We fed tadpoles *ad libitum* with pulverized frog brittle (NASCO, Fort Atkinson, WI) or Sera Micron plankton food. Tadpoles were staged using the developmental staging system of Nieuwkoop and Faber [26] (NF). Animals were euthanized following humane methods to alleviate suffering by immersion in 0.1% benzocaine followed by decapitation. All procedures involving animals were conducted under an approved animal use protocol (PRO00006809) in accordance with the guidelines of the Institutional Animal Care and Use Committee at the University of Michigan.

### RNA sequencing (RNA-seq)

The RNA-seq datasets used for the current analyses were previously published [21,25] and the data deposited in GEO (developmental: GSE139267; 3,5,3'-triiodothyronine ($T_3$)-induced: GSE130816). For the developmental gene expression analysis, we measured mRNA levels in tadpole preoptic area/thalamus/hypothalamus (S1 Fig) at four stages of metamorphosis (NF50: premetamorphosis, when the larvae grow but little or no morphological change occurs and plasma TH concentrations are low; NF56: prometamorphosis, when hindlimb growth accelerates and plasma TH concentration rises; NF62: metamorphic climax, the most rapid phase of morphological change when thyroid activity is at its peak; NF66: completion of metamorphosis, juvenile frog; n = 3/developmental stage) [25]. For analysis of $T_3$-induced gene expression changes we measured mRNA by RNA-seq in preoptic area/thalamus/hypothalamus of premetamorphic tadpoles (NF54) treated with or without $T_3$ (5 nM dissolved in the aquarium water) for 16 h (n = 3/treatment) [21]. We re-analyzed the two RNA-seq datasets with the same bioinformatics tools as before [21,25], but using the most recent *X. tropicalis* genome build (v9.1) from Ensembl and setting a false discovery rate (FDR) adjusted p value <0.05. We generated heatmaps for the top 100 upregulated and the top 100 downregulated genes determined during the developmental interval NF50 to NF62 using the software Heatmapper [27]. We clustered the genes in the developmental RNA-seq dataset into groups based on their expression profiles using the software *Clust* [28]. We used gProfiler software [29] to conduct functional enrichment analysis of gene ontology (GO), KEGG pathway, and reactome (REACT) pathways within the set of differentially expressed genes for each comparison. KEGG pathway data were rendered using Pathview software [30].

### Reverse transcriptase real time quantitative PCR (RTqPCR)

Changes in gene expression discovered by RNA-seq were previously validated by analysis of a subset of both up and down-regulated genes using RTqPCR [21,25]. Here we analyzed the mRNA levels of a subset of cell cycle control genes using RTqPCR following previously published methods [25]. Oligonucleotide primers used for RTqPCR are given in Table 1.

**Table 1. Oligonucleotide primers used for RTqPCR and chromatin immunoprecipitation (ChIP) assay.**

| RTqPCR | |
| --- | --- |
| Gene | Primer sequence (5' → 3') |
| ccnb1.2 | Fwd: GAGGATGCACAAGCAGTCAG<br>Rev: TTTGGGAACTGGGTGTTCCT |
| ccna2 | Fwd: TTTGACCTTGCTGCTCCAAC<br>Rev: CGCAGGAAAGGATCAGCATC |
| cdk1 | Fwd: GGAACGCCCAACAATGAAGT<br>Rev: CAGGTCCAGCCCATCCTTAT |
| e2f1 | Fwd: CGCTGACGTTGTGTATGGTT<br>Rev: TTCCTGTAGGCATTCACGGT |
| a-actinin (reference gene) | Fwd: GGACAATTATCCTGCGTTTTGC<br>Rev: CCTTCTTTGGCAGATGTTTCTTC |
| **ChIP assay** | |
| Genomic region | Primer sequence (5' → 3') |
| thrb TRE | Fwd: CCCCTATCCTTGTTCGTCCTC<br>Rev: GCGCTGGGCTGTCCT |
| klf9 synergy module (KSM) | Fwd: CCGTCCCTTCTTTTGTGTACATT<br>Rev: GCTGTTCGTGCCACTTTGC |
| thibz TRE | Fwd: GGACGCACTAGGGTTAAGTAAGG<br>Rev: TCTCCCAACCCTACAGAGTTCAA |
| ifabp promoter | Fwd: CCCTACATTGGTTGAGCCAGTTTT<br>Rev: TCAAAGGCCATGGTGATTGGT |
| thrb exon 5 | Fwd: CCCCGAAAGTGAAACTCTAACTCT<br>Rev: CCACACCGAGTCCTCCATTTT |

## Thyroid hormone receptor (TR) chromatin immunoprecipitation (ChIP) and TR ChIP sequencing (TR ChIP-seq)

We chose metamorphic climax (NF62) to conduct TR ChIP-seq on tadpole brain because this stage corresponds to the highest circulating plasma TH concentration and *thyroid hormone receptor alpha* (*thra*) and *thyroid hormone receptor beta* (*thrb*) mRNA levels [1,31]. Also, this stage is when the expression of most regulated genes is highest or lowest, and the rate of tissue transformation is maximal, reflecting active transcriptional regulation by liganded TRs [1]. We isolated chromatin from whole brain (NF62; 5 brains pooled per replicate) and conducted targeted TR ChIP and TR ChIP-seq following our previously published methods [32]. Briefly, we cross-linked the chromatin with formaldehyde, fragmented it by sonication, then precipitated it (5 μg per reaction) using a polyclonal antiserum to *Xenopus* TRs (PB antiserum provided by Yun-Bo Shi) and the Magna ChIP A/G kit (Millipore) following the manufacturer's protocol. We analyzed eight replicate ChIP DNA samples using qPCR assays targeting three positive control regions (the *Krüppel-like factor 9—klf9* synergy module; the *thrb* TH response element–TRE; and the *thyroid hormone induced bZip protein–thibz—*TRE) and two negative control regions (the *intestinal fatty acid binding protein* [*ifabp*] promoter and *thrb* exon 5), then we selected and pooled three replicates with the highest signal/noise ratio. We submitted the sample to the University of Michigan DNA sequencing core along with the input sample for library preparation using the SMARTer ThruPLEX DNA-Seq Kit (Takara), and sequencing in a single lane of an Illumina 4000 Hi-Seq machine at the University of Michigan DNA Sequencing Core.

We conducted quality processing on raw reads (fastq) files from each sample to remove poor quality bases and adapter sequences using fastp (v0.20.1). Quality assessments on raw and trimmed reads were performed using FastQC (v0.11.9). We aligned processed reads to the *X.*

*tropicalis* genome (v9.1), Ensembl release 101, using bwamem (v0.7.17) and default parameters. Alignments were then sorted, indexed using samtools (v1.13), and duplicates marked using Picard (v2.18.2). We filtered unmapped reads, secondary alignments, PCR or optical duplicates, and mappings with a quality below 30 with samtools (v1.13). We created read depth-normalized bigwig files from sorted files using bamCoverage from deepTools (v3.5.1). We identified areas of enriched TR binding using MACS2 (v2.2.6) with the sharp peak type. The regions of the *X. tropicalis* genome where TR associates in chromatin were visualized using the integrative genome viewer (IGV). For binding sites within 10 kb upstream of a transcription start site (TSS) or within a gene-body, we used ChIPPeakAnno (v3.26.4) to annotate peaks with the corresponding gene name; the distance from the peak to the gene; and the relative position of the peak relative to the gene (ie., upstream vs. internal). We then plotted the genomic distribution of TR binding sites across genomic elements using ChIPPeakAnno (v3.26.4) and ChIPseeker (v1.28.3). We used ChIPseeker (v1.28.3) to create scaled (3 kb upstream and downstream) profile plots of the TR binding sites relative to the TSS and gene-body.

Differentially expressed genes (DEGs) were annotated with TR binding site information, defined as enriched peaks with a p-value ≤0.0005, if the peaks were within 10 kb upstream of the gene TSS or within the gene-body. Output from different quality control tools and the peak calling were summarized with MultiQC (v1.11). Select parts of the TR ChIP-seq dataset were previously reported, which included validations using targeted ChIPqPCR assays using chromatin preparations distinct from the chromatin used for ChIP-seq at six known TH-TR target genes (*thrb*, *thibz*, *klf9*, *growth arrest and DNA damage inducible gamma—gadd45g*, *ten eleven translocase 2—tet2*, and *tet3*) and two negative control regions (*ifabp* promoter, *thrb* exon 5) [32]. We also conducted targeted ChIPqPCR assays to validate TR ChIP-seq peaks at three uncharacterized loci (S2 Fig). The TR ChIP-seq dataset has been deposited in GEO (GSE214697).

## Statistical analysis

We analyzed RTqPCR data using SYSTAT (v. 13; Systat Software, San Jose, CA). We used one-way ANOVA followed by Fisher's least significant difference (Fisher's LSD) *post hoc* test (a = 0.05). Derived values were $\text{Log}_{10}$-transformed before statistical analysis if the variances were found to be heterogeneous.

## Results

### RNA-sequencing analysis in tadpole brain during spontaneous metamorphosis

We conducted pairwise analysis of gene expression changes in tadpole brain for five developmental intervals: premetamorphosis to prometamorphosis (NF50-56; i.e., early prometamorphosis); prometamorphosis to metamorphic climax (NF56-62; i.e., late prometamorphosis); metamorphic climax to the completion of metamorphosis (NF62-66); premetamorphosis to metamorphic climax (NF50-62); and premetamorphosis to the completion of metamorphosis (NF50-66) (Tables 2 and S1). There were 5561 unique genes whose mRNA level changed in one or more of the developmental interval comparisons, which represents 26% of the protein coding genes in *X. tropicalis* (based on an estimated 21,000 genes) [33]. The numbers of differentially expressed genes (DEGs) with pairwise developmental stage comparisons are given in Fig 1A and Tables 2 and S1. We plotted the RNA-seq counts for four known $T_3$-regulated genes and found the expected changes during metamorphosis (Fig 1B). Heatmaps depicting expression changes during metamorphosis

**Table 2. Top twenty genes with annotation upregulated from premetamorphosis (NF50) to metamorphic climax (NF62) in tadpole brain.**

| Gene symbol | Gene title | ENSEMBL | Log2fold change* | Adjusted p value |
|---|---|---|---|---|
| dio3 | deiodinase, iodothyronine, type 3 | ENSXETG00000036613 | 3.275125254 | 2.94E-40 |
| avt | arginine vasotocin** | ENSXETG00000017979 | 2.328280168 | 1.28E-20 |
| thibz | thyroid hormone induced bZip protein | ENSXETG00000007922 | 2.296353968 | 1.19E-09 |
| spock2 | indolethylamine N-methyltransferase | ENSXETG00000026030 | 2.28515105 | 4.84E-12 |
| vstm5 | V-set and transmembrane domain containing 5 | ENSXETG00000017117 | 2.266298133 | 5.32E-29 |
| phf21a | BRAF35-HDAC complex protein BHC80 | ENSXETG00000011757 | 2.214433985 | 2.69E-32 |
| inhbb | inhibin subunit beta B | ENSXETG00000036045 | 2.200045605 | 1.99E-09 |
| pmepa1 | prostate transmembrane protein, androgen induced 1 | ENSXETG00000011263 | 2.195093687 | 1.46E-24 |
| apoe | apolipoprotein E | ENSXETG00000035511 | 2.127879577 | 8.93E-31 |
| dao | D-amino-acid oxidase | ENSXETG00000022308 | 2.112579694 | 3.37E-24 |
| cyp27a1 | cytochrome P450 family 27 subfamily A member 1 | ENSXETG00000025024 | 2.083714291 | 1.02E-08 |
| p2rx5 | purinergic receptor P2X, ligand gated ion channel, 5 | ENSXETG00000032944 | 1.98969067 | 2.89E-10 |
| cela1.2 | chymotrypsin like elastase 1, gene 2 | ENSXETG00000039717 | 1.976354459 | 2.54E-07 |
| nlrx1 | NLR family member X1 | ENSXETG00000021586 | 1.949234863 | 5.22E-36 |
| arhgap24 | Rho GTPase activating protein 24 | ENSXETG00000006200 | 1.887018231 | 7.6E-09 |
| bhlhe41 | basic helix-loop-helix family member e41 | ENSXETG00000023482 | 1.884565228 | 5.58E-17 |
| arhgap45 | Rho GTPase activating protein 45 | ENSXETG00000011260 | 1.877904 | 6.66E-22 |
| chrm2 | cholinergic eceptor muscarinic 2 | ENSXETG00000030443 | 1.877869602 | 1.87E-12 |
| sbf2 | zinc finger protein 816 | ENSXETG00000039440 | 1.825679583 | 4.21E-11 |
| trpm8 | transient receptor potential cation channel, subfamily M, member 8 | ENSXETG00000026873 | 1.824899962 | 3.94E-15 |

* Log2fold change of 1 = 2 fold increase.

** This gene is mislabeled in the genome database as *arginine vasopressin* (*avp*) which is a mammalian gene. The amphibian gene is *arginine vasotocin* (*avt*).

for the top 100 upregulated and the top 100 downregulated genes (determined during the developmental interval NF50-NF62) are shown in Fig 1C.

The top 20 genes with annotation that were upregulated from NF50 to NF62 are listed in Table 2, and the top 20 genes downregulated during this interval are listed in Table 3. Not included in Table 2 are two hemoglobin subunit genes (*hba1* and *hbg1*) that were strongly upregulated during metamorphosis, and are likely derived from residual blood remaining in the brain at the time of harvest. During amphibian metamorphosis there is a switch from larval to adult erythrocytes that express different globin genes, with *hba1* and *hbg1* expressed in adult but not in larval cells [40,41].

**Patterns of gene regulation during metamorphosis.** Clustering analysis revealed five major patterns (C1 –C5) of gene regulation during metamorphosis (Fig 2A). Two clusters (C1 and C2) include genes that were downregulated during metamorphosis. The C1 genes, which represent 34.4% of all genes regulated during metamorphosis, were downregulated from NF50 to NF62, but trended up from NF62 to NF66. The C2 genes, which represent 5.5% of all regulated genes, were downregulated from NF50 to NF62 and remained low at NF66. Two clusters (C4 and C5) include genes that were upregulated during metamorphosis. The C4 genes, which represent 34.1% of all genes regulated during metamorphosis, were upregulated from NF50 to NF62, but trended down slightly from NF62 to NF66. The C5 genes, which represent 17.1% of all genes regulated during metamorphosis, were upregulated from NF50 to NF62 and remained elevated at NF66. The C3 genes, which represent 8.9% of all genes regulated during metamorphosis, include genes that were strongly upregulated from NF50 to NF56, remained elevated NF56 to NF62, but then were downregulated from NF62 to NF66.

Pair-wise comparisons of gene expression changes at four stages of metamorphosis

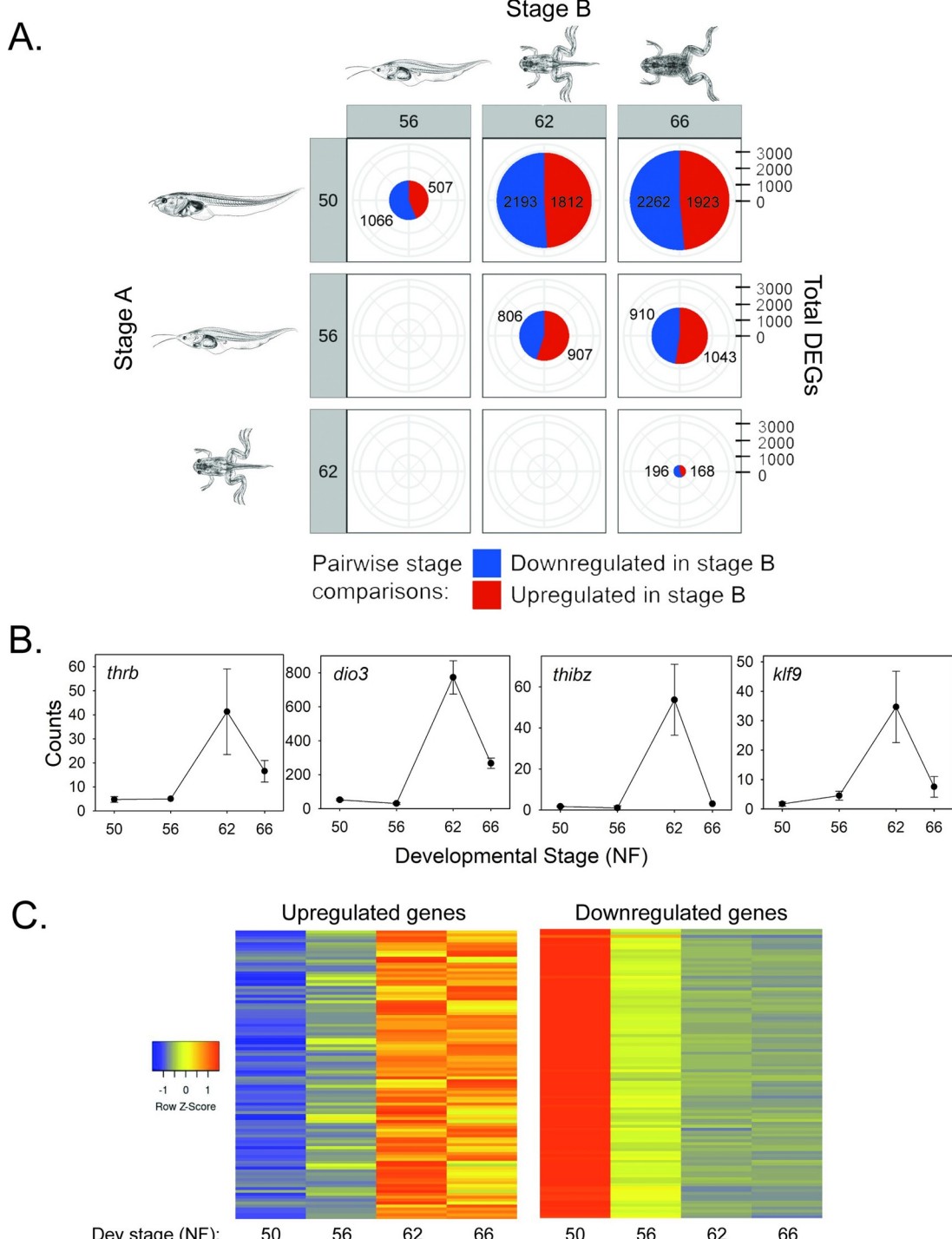

**Fig 1. RNA-sequencing analysis in tadpole brain during metamorphosis. A.** Pie charts with pairwise comparisons of differentially expressed genes (DEGs) between four stages of metamorphosis. Numbers in red areas of the pie charts represent genes upregulated in Stage B, while the numbers in the blue areas of the pie charts represent genes downregulated in Stage B. Gene regulation changes increased as metamorphosis progressed, with roughly half the genes upregulated and half downregulated. *Xenopus* illustrations © Natalya Zahn (2022), source Xenbase (www.xenbase.org RRID:SCR_003280) [34]. **B.** Expression patterns of four known TH-regulated genes in tadpole brain (preoptic area/thalamus/hypothalamus) analyzed at four Nieuwkoop-Faber (NF) stages of spontaneous metamorphosis. Plotted are data from an RNA-seq experiment (n = 3/NF developmental stage). *thrb–thyroid hormone receptor b*; *dio3 –monodiodinase type 3*; *thibz–thyroid hormone induced bZip protein*; *klf9 –Krüppel-like factor 9*. All except *dio3* have been shown to be directly regulated by TH-TR [35–39], which is also supported by the current TR ChIP-seq experiment. **C.** Heatmaps showing expression changes for the top 100 upregulated and the top 100 downregulated genes (determined during the developmental interval NF50-NF62; S1 Table).

These data are summarized in Table 4.

**Early prometamorphosis (NF50-56):** We identified 1573 genes whose mRNAs levels changed during this interval, with gene downregulation predominating (32% increased, 68% decreased). This developmental period represents the initiation of metamorphosis, and is characterized by rising plasma TH titers, and external changes like the growth and differentiation of the hindlimbs.

**Late prometamorphosis (NF56-62):** We identified 1713 genes whose mRNAs levels changed during this interval, with similar numbers up- and downregulated (53% increased, 47% decreased). This developmental period is characterized by sharply increasing plasma TH titers that peak at metamorphic climax, and external changes like continued hindlimb growth and differentiation, forelimb emergence and growth, intestinal remodeling, cranial re-structuring and the beginning of gill and tail resorption.

**Metamorphic climax to the completion of metamorphosis (NF62-66):** We identified 364 genes whose mRNA levels changed during this interval, 46% up- and 54% downregulated. This developmental period is characterized by a sharp decline in plasma TH titers, completion of resorption of the tail and gills, cranial restructuring, and the emergence of the juvenile frog.

**Premetamorphosis to metamorphic climax (NF50-62):** During the entire period encompassing premetamorphosis to the climax of metamorphosis there were 4005 genes whose mRNA levels changed, 45% up- and 55% downregulated. This developmental period is characterized by a large increase in plasma TH titers from nondetectable to peak concentration, and the progression of the morphological changes that characterize the transformation of the premetamorphic tadpole into the juvenile frog.

**Table 3. Top twenty genes with annotation downregulated from premetamorphosis (NF50) to metamorphic climax (NF62) in tadpole brain.**

| Gene symbol | Gene title | ENSEMBL | Log2fold change* | Adjusted p value |
|---|---|---|---|---|
| pla2g2e | phospholipase A2 group IIE | ENSXETG00000037667 | -4.444463949 | 3.8805E-38 |
| col6a6 | ribonucleotide reductase M2, gene 2 | ENSXETG00000036052 | -4.106238088 | 5.074E-114 |
| crisp1.7 | cysteine-rich secretory protein 1 gene 7 | ENSXETG00000011152 | -3.986829953 | 1.9676E-29 |
| cdca7 | cell division cycle associated 7 | ENSXETG00000007518 | -3.541269862 | 1.1933E-55 |
| cct6a | component of the chaperonin-containing T-complex (TRiC) | ENSXETG00000006636 | -3.498033622 | 1.8951E-53 |
| sapcd1 | suppressor APC domain containing 1 | ENSXETG00000041061 | -3.463026967 | 4.1714E-66 |
| eef1a1o | eukaryotic translation elongation factor 1 alpha 1, oocyte form | ENSXETG00000022604 | -3.330857635 | 1.5064E-59 |
| neurog1 | neurogenin 1 | ENSXETG00000007898 | -3.307301672 | 1.6585E-31 |
| smc2 | structural maintenance of chromosomes 2 | ENSXETG00000010209 | -3.307001586 | 1.9573E-74 |
| pbk | PDZ binding kinase | ENSXETG00000020438 | -3.236977659 | 2.3562E-47 |
| kif20a | kinesin family member 20A | ENSXETG00000006721 | -3.193016724 | 2.1544E-40 |
| cdk2 | cyclin-dependent kinase 2 | ENSXETG00000009444 | -3.185053334 | 1.8627E-44 |
| cdk1 | cyclin-dependent kinase 1 | ENSXETG00000003123 | -3.183432448 | 4.228E-56 |
| nusap1 | nucleolar and spindle associated protein 1 | ENSXETG00000027499 | -3.154574577 | 3.32E-68 |
| kif23 | kinesin family member 23 | ENSXETG00000007829 | -3.15195982 | 2.6852E-47 |
| cdca8 | cell division cycle associated 8 | ENSXETG00000002167 | -3.1382786 | 4.1909E-31 |
| ccna2 | cyclin A2 | ENSXETG00000001016 | -3.121308956 | 6.1124E-44 |
| dlc | putative ortholog of delta-like protein C precursor | ENSXETG00000002875 | -3.090104341 | 1.629E-79 |
| atad2 | ATPase family, AAA domain containing 2 | ENSXETG00000023215 | -3.068914576 | 3.5354E-28 |
| hmmr | hyaluronan-mediated motility receptor (RHAMM) | ENSXETG00000007906 | -3.065327172 | 5.3941E-46 |

* Log2fold change of -1 = 50% decrease.

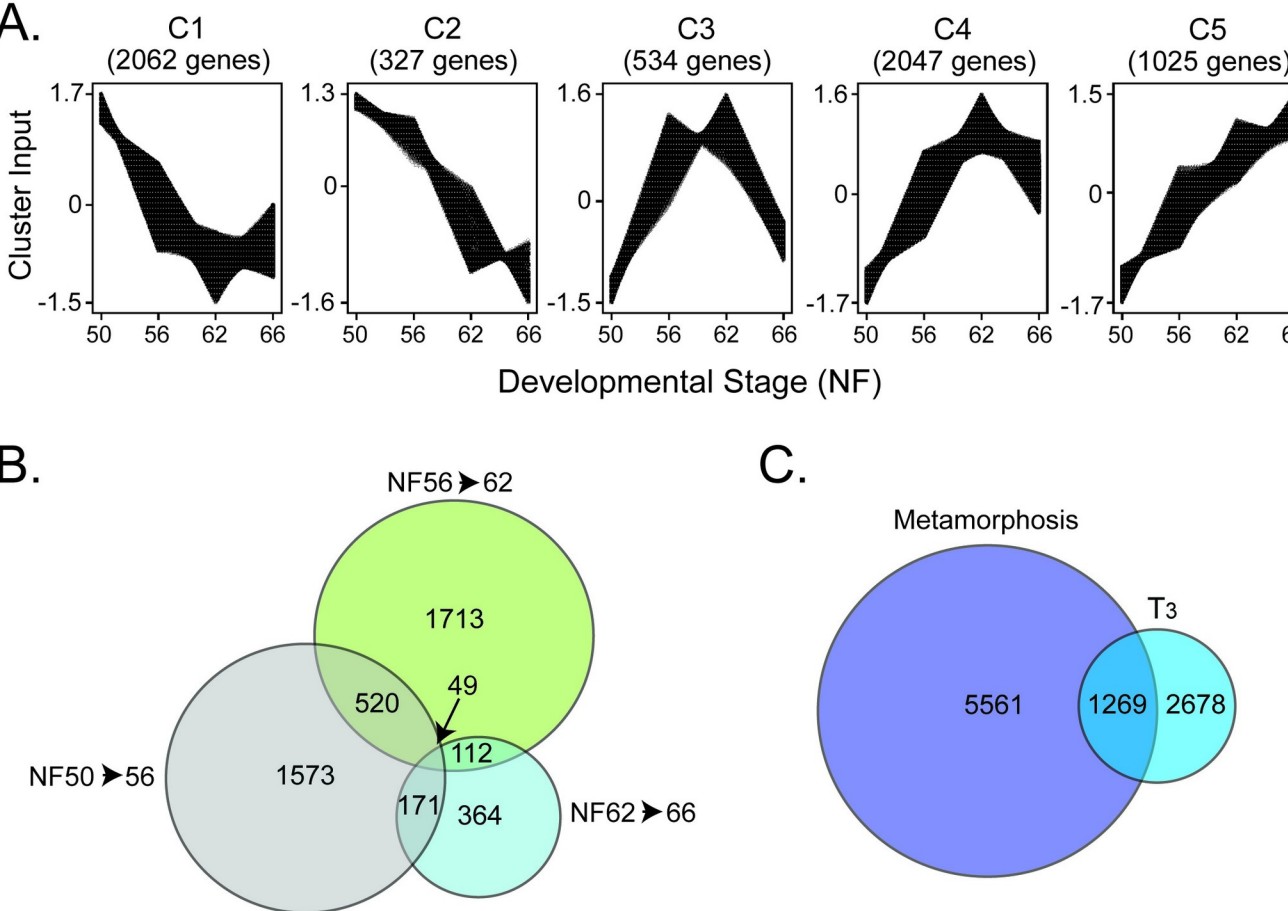

**Fig 2. Patterns of gene regulation in *X. tropicalis* tadpole brain during spontaneous metamorphosis or after T₃ treatment analyzed by RNA-sequencing. A.** Clustering analysis showing five patterns of gene expression changes across four stages of metamorphosis. **B.** Venn diagram showing numbers of genes regulated with overlaps in three developmental stage comparisons. C. Venn diagram showing the overlap between all genes that changed expression during spontaneous metamorphosis with genes induced or repressed by exogenous T3 in premetamorphic tadpoles.

**Premetamorphosis to the completion of metamorphosis (NF50-66):** We identified 4185 genes whose mRNA levels changed during the entire interval from the beginning to the end of metamorphosis, with 46% up- and 54% downregulated.

**Overlap:** We also looked at the numbers of DEGs that were common to pairs of selected developmental intervals. Comparing the developmental intervals NF50-56 and NF56-62, there were 520 common DEGs, which represented 33.1% and 30.4%, respectively, of all genes regulated during these intervals (Fig 2B and S2 Table). Thus, about one third of the genes whose mRNA levels changed during early prometamorphosis also showed changes during late prometamorphosis/climax; i.e., the mRNA levels for these genes continued to increase or decrease as metamorphosis proceeded to climax.

Comparing the developmental intervals NF56-62 and NF62-66, there were 171 common DEGs, which represented 10% and 47%, respectively, of all genes regulated during these intervals (Fig 2B). Thus, only one tenth of the genes whose mRNA levels changed during late prometamorphosis/climax also showed changes from climax to the completion of metamorphosis. Comparing the developmental intervals NF50-56 and NF62-66, there were 112 common DEGs, which represented 7.1% and 30.8%, respectively, of all genes regulated

**Table 4. Comparisons of gene counts from RNA-sequencing experiments.**

| Comparison | Numbers of differentially expressed genes | | |
| --- | --- | --- | --- |
| | All | Up | Down |
| Premetamorphosis (NF50) vs prometamorphosis (NF56) | 1573 | 507 | 1066 |
| Prometamorphosis (NF56) vs metamorphic climax (NF62) | 1713 | 907 | 806 |
| Metamorphic climax (NF62) vs metamorphic frog (NF66) | 364 | 168 | 196 |
| Prometamorphosis (NF56) vs metamorphic frog (NF66) | 1953 | 1043 | 910 |
| Premetamorphosis (NF50) vs metamorphic climax (NF62) | 4005 | 1812 | 2193 |
| Premetamorphosis (NF50) vs metamorphic frog (NF66) | 4185 | 1923 | 2262 |
| Number of unique genes regulated during metamorphosis* | 5561 | | |
| $T_3$ treatment of premetamorphic (NF50) tadpoles | 2678 | 1396 | 1282 |
| Number of unique genes whose mRNA levels change during metamorphosis plus after $T_3$ treatment of premetamorphic tadpoles | 6957 | | |
| Number of genes whose mRNA levels change during metamorphosis and after $T_3$ treatment of premetamorphic tadpoles | 1269 | | |
| Number of genes whose mRNA levels are regulated by $T_3$ in premetamorphic tadpoles but do not change during metamorphosis | 1409 | | |
| Number of genes whose mRNA levels change during metamorphosis but are not regulated by $T_3$ in premetamorphic tadpoles | 4297 | | |

*These are the unique genes that change their mRNA levels in one or more of the pair-wise developmental interval comparisons.

See also S1 and S2 Tables for a full accounting of the RNA-seq data.

during these intervals (Fig 2B). There were 49 genes that were common between the three developmental intervals. These comparisons show that the expression level of most genes at metamorphic climax, high or low, is maintained in the juvenile frog.

## RNA-sequencing analysis in brain of premetamorphic tadpoles treated with $T_3$

Treatment of premetamorphic tadpoles (NF54) with $T_3$ (5 nM) for 16 h caused statistically significant changes in the mRNA levels of 2678 genes in tadpole brain (preoptic area/thalamus/hypothalamus); 1396 (52%) were induced and 1282 (48%) were repressed (Fig 2C and Tables 2 and S1). This number represents 13% of the protein coding genes in *X. tropicalis*. These numbers differ slightly from our previously published estimates [21] owing to our use of the latest genome build in the current study.

## Overlap of the developmental and $T_3$-induced gene regulation programs

Of the unique genes whose mRNA levels changed during the entire metamorphic period (NF50 to NF66; 5561 genes), 1269 genes (22.82%) were common with genes regulated by exogenous $T_3$ in premetamorphic tadpole brain (Fig 2C and Tables 2 and S3). This percentage was similar (22.5%) when comparing the interval NF50-56 when circulating TH concentration is low, but greater (895 of 4005 genes = 33.42%) for the interval NF50-62 when circulating TH concentration increases and peaks (Fig 2B and Tables 2 and S2). Thus, 77% of the genes expressed in tadpole brain whose mRNA levels changed during the entire metamorphic period, and 66% of the genes that changed during NF50-62 were not modulated by exogenous $T_3$ in premetamorphic tadpole brain.

On the other hand, of the genes that were regulated by exogenous $T_3$ in premetamorphic tadpole brain (2678 genes), the mRNA levels of 47.39% of these changed during spontaneous

metamorphosis. Thus, over half of the genes that are regulated by exogenous $T_3$ in premetamorphic tadpole brain do not change their mRNA levels during metamorphosis.

We also analyzed the direction of change in gene expression, up- or downregulated, for the genes that overlapped between the developmental intervals and the $T_3$-induced genes. The concordance was 49.2% for NF50-56, 81% for NF56-62, and 67.2% for NF50-62 (S3 Table). Thus, depending on the developmental interval analyzed, ~20% to 50% of the genes that change their expression during spontaneous metamorphosis do so in a direction opposite to that caused by treatment of premetamorphic tadpoles with $T_3$.

## TR association in chromatin in tadpole brain identified by ChIP-sequencing

We identified 6302 unique peaks by TR ChIP-seq analysis of brain chromatin from metamorphic climax stage (NF62) tadpoles (hereafter 'TR peaks'; Tables 5 and S4). The peaks were broadly distributed across each of the 10 *X. tropicalis* chromosomes (Fig 3A). Our prior validation of the TR ChIP-seq data using targeted ChIP assay confirmed TR association in chromatin at the TRE regions of *thrb*, *thibz*, *klf9*, *gadd45g*, *tet2* and *tet3*; conversely, there were no TR peaks at two negative control regions (*ifabp* promoter and *thrb* exon 5) [32]. Notably, the immediate early gene *klf9*, whose mRNA level increases at the onset of metamorphosis and shows rapid induction kinetics in response to exogenous $T_3$ in premetamorphic tadpoles [38,42], exhibited a large TR peak at the previously characterized and evolutionarily conserved TRE located in an upstream enhancer (the *klf9* synergy module–KSM—located ~6 kb upstream of the TSS; S3 Fig) [39]. We also discovered additional, previously uncharacterized TR peaks at *klf9*: one farther upstream to the TSS (~ -7kb), one just upstream of the TSS, one overlapping the TSS, and one within the gene body (S3 Fig and S5 Table).

**Table 5. Accounting of TR ChIP-seq peaks.**

| TR ChIP-seq dataset | |
|---|---|
| **Comparison** | **Number** |
| Total number of TR peaks | 6302 |
| Number of TR peaks associated with genes | 5098 |
| Number of genes* with associated TR peaks | 6512 |
| Number of genes without TR peaks | 13783 |
| Number of TR peaks with two or more genes | 1546 |
| Number of genes with multiple TR peaks | 921 |
| Number TR peaks not associated genes | 1336 |
| Percentage of TR peaks associated with genes | 80.89% |
| **TR ChIP-seq peaks at differentially expressed genes (DEGs) and non-DEGs** | |
| Number of DEGs from beginning to the end of metamorphosis with TR peaks | 1355 |
| Number of DEGs from premetamorphosis (NF50) to metamorphic climax (NF62) with multiple TR peaks | 207 |
| Number of DEGs from beginning to the end of metamorphosis without TR peaks | 4206 |
| Number of non-DEGs from beginning to the end of metamorphosis with TR peaks | 3743 |
| Number of DEGs after $T_3$ treatment of premetamorphic (NF50) tadpoles with TR peaks | 703 |
| Number of DEGs after $T_3$ treatment of premetamorphic (NF50) tadpoles without TR peaks | 1975 |

\* Some TR peaks are associated with more than one gene, and some genes have mulitple TR peaks

See S4, S5 and S6 Tables for a full accounting of the TR ChIP-seq data.

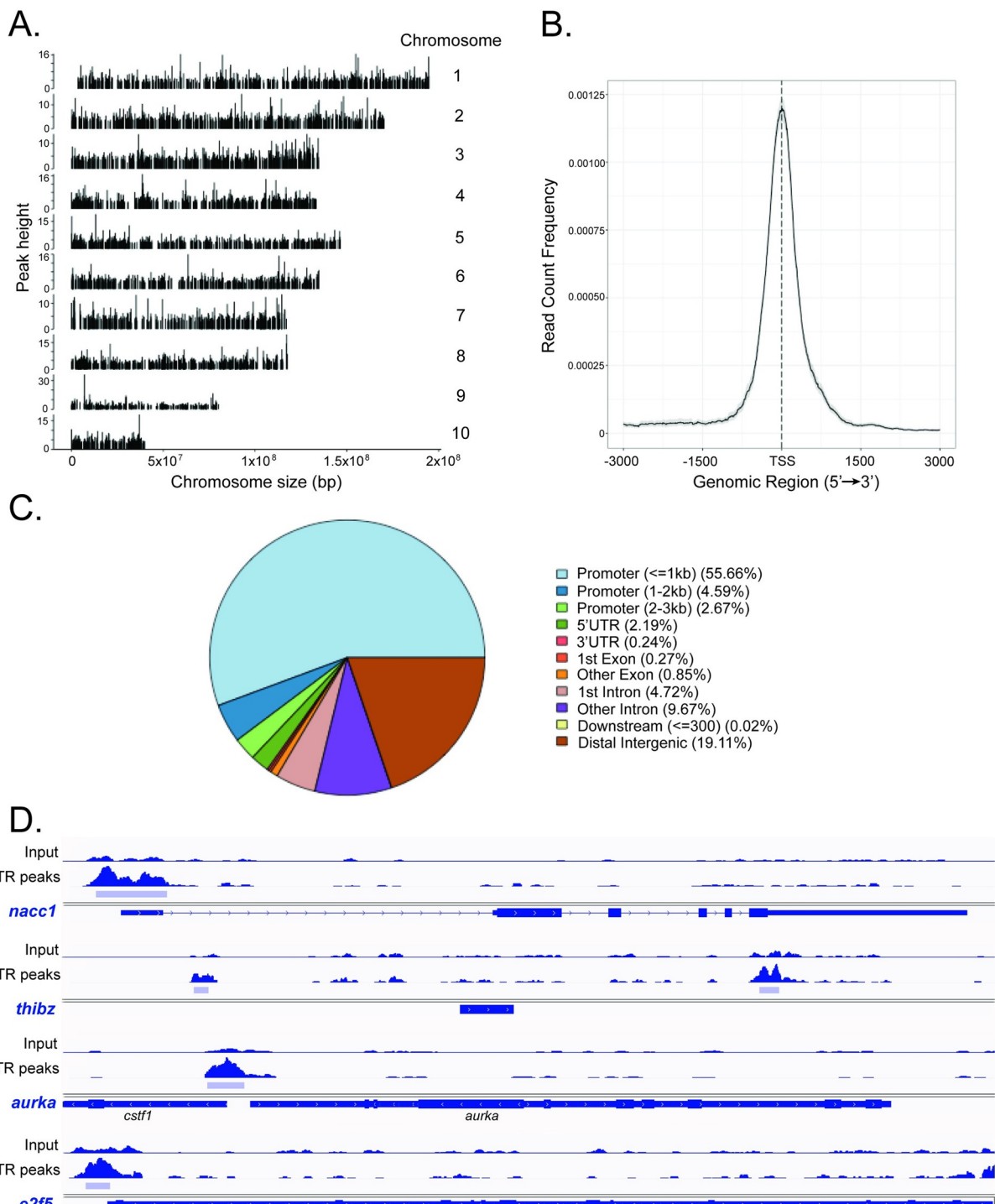

**Fig 3. Distribution across the genome of thyroid hormone receptor ChIP-seq peaks in *X. tropicalis* neural cells. A.** TR ChIP-seq peaks were found across the entire genome and uniformly distributed on the 10 chromosomes of *X. tropicalis*. **B.** A majority of TR ChIP-seq peaks were located +/- 1 kb from the transcription start sites (TSS) of genes. **C.** Pie chart showing the distribution of TR ChIP-seq peaks across the *X. tropicalis* genome by genomic feature. **D.** IGV genome browser tracks showing the locations of TR ChIP-seq peaks (TR peaks; light blue bars) at four loci. Shown are two genes that were upregulated (*nucleus accumbens associated 1—nacc1*, peak range 0–291; *thyroid hormone induced bZip protein—thibz*, peak range 0–250) and 2 that were downregulated (*aurora kinase A—aurka*, peak range 0–326; *E2F transcription factor 5—e2f5*, peak range 0–233) during metamorphosis. Gene structures are shown in dark blue below the genome tracks.

By setting the window of discovery 10 kb upstream of TSSs and within gene bodies, we found a total of 5098 TR peaks associated with unique genes (which represented 31.29% of the 18881 genes included in the dataset; Tables 5 and S5). Thus, most of the total unique TR peaks (5098 of 6302; 80.89%) were associated with genes using this discovery window. Some TR peaks were associated with more than one gene (1546 peaks), and some genes had multiple TR peaks (921 genes; Tables 5 and S5 and S6). A majority of the TR peaks were clustered +/- 1 kb from the TSSs of the associated genes (Fig 3B); 62.92% were found 0–3 kb upstream, 55.66% </ = 1 kb from TSSs, and 17.3% within gene bodies with 13.75% located within introns (Fig 3C). Four genome traces with TR peaks at two upregulated and two downregulated genes are shown in Fig 3D.

Using HOMER software we found that the direct repeat with 4 base spacer (DR4) THRa NR and THRb NR motifs were enriched in the DNA sequences within the TR peaks (S7 Table), similar to the findings of Tanizaki and colleagues [43] who recently conducted TR ChIP-seq on tadpole intestine.

**TR peaks and DEGs.** We looked at the association of TR peaks with DEGs during metamorphosis, or after $T_3$ treatment of premetamorphic tadpoles. Of the 5561 genes whose mRNA levels changed in tadpole brain during spontaneous metamorphosis, 1355 genes (24.4%) had TR peaks; 644 of these genes were upregulated and 711 downregulated during metamorphosis (NF50-NF66; S6 Table). There were 1773 TR peaks associated with these DEGs (some genes had multiple TR peaks) which represents 34.8% of all TR peaks that were associated with genes (5098); 885 peaks were associated with upregulated, and 888 with downregulated genes.

There were 703 TR peaks associated with genes that were induced or repressed by $T_3$ in premetamorphic tadpole brain (S6 Table). Of the 2678 genes whose mRNA levels were modulated by $T_3$ treatment, 269 (10%) had TR peaks: 179 of the induced and 90 of the repressed genes (some genes had multiple TR peaks). Of the 6970 unique DEGs from the combined developmental and $T_3$-treated datasets, 1624 (23.3%) had TR peaks.

**Major cellular pathways regulated during metamorphosis or after $T_3$ treatment of premetamorphic tadpoles.** We used gProfiler to analyze gene ontology (GO) terms, KEGG and Reactome (REACT) cellular pathways represented by genes within each of the gene regulation clusters (S8 Table) and by developmental interval (S9 Table). Genes of C1 and C2, which were downregulated during metamorphosis, comprise cellular pathways involved with DNA repair, DNA replication, cell cycle and protein synthesis (ribosome biogenesis, protein translation). Genes of C4 and C5, which were upregulated and remained elevated at the completion of metamorphosis, comprise a diversity of cellular signaling pathways involved with neural cell structure and function, neural signaling and neuroendocrine function,. Genes of C3, which were first up-, then downregulated during metamorphosis, comprise a limited number of cellular processes that included protein kinase and transcriptional coregulator activity, and cellular protein modification. The major cellular processes influenced by $T_3$ treatment were RNA metabolism and protein synthesis (S10 Table).

**Gene expression patterns during metamorphosis.** We looked at the DEGs in the different clusters to better understand specific cellular processes occurring during metamorphosis (S11 Table). We also determined if the DEGs had TR peaks (S5 and S6 Tables).

**Cell cycle:** There was a broad downregulation of genes involved with cell cycle and cell division in tadpole brain during metamorphosis. These included *myc proto-oncogene* (myc), two *aurora kinase (aurk)*, nine *cyclin (ccn)*, 14 *cell division cycle (cdc)*, five *cyclin-dependent kinase* (*cdk*) and five *E2F transcription factor* (*e2f*) genes, among others. Genes involved with cell cycle regulation were among the most strongly downregulated, from the beginning (NF50) to

the end (NF66) of metamorphosis (e.g., log2FoldChange: *cdk1* = -4.49; *ccnb1.2* = -4.19; *ccna2* = -4.13; *cdca7* = -3.88; *cdk2* = -3.83; *ccnb3* = -3.79; among others–see S11 Table).

Using targeted RTqPCR we confirmed the developmental decline in the mRNA levels of four cell cycle genes that are known targets of E2F transcription factors: *ccnb1.2*, *ccna2*, *cdk1* and *e2f1*. Shown for comparison in Fig 4A are the RNA-seq count and the RTqPCR data, which both show large declines in the mRNA levels of these four genes from NF50 through the completion of metamorphosis. For the RTqPCR experiment we used different tissue samples from the RNA-seq experiment and we included NF54 and NF58 (the RNA-seq experiment had NF56); this showed that the decline in the mRNA levels for these genes began after NF 54 or later. The KEGG cell cycle pathway is shown in Fig 4B. Several downregulated genes involved with cell cycle/cell division including *myc*, *aurka*, *ccnf*, three *e2f* and four *cdc* genes, among others, had TR peaks.

The Wnt/b-catenin signaling pathway is implicated in TH-dependent cell proliferation [44,45]. We saw a broad downregulation of Wnt/b-catenin pathway genes in tadpole brain during metamorphosis, including five *Wnt family member*, *wntless*, *Wnt ligand secretion mediator*, 4 *frizzled class receptor* (one with TR peaks) and four *secreted frizzled-related protein* genes (see Fig 4C for the RNA-seq expression profiles of four of these genes).

Earlier, using bromodeoxyuridine labeling, we saw a large increase in cell proliferation in tadpole brain during early prometamorphosis that was dependent on endogenous TH, then the rate of proliferation declined through metamorphic climax [46]. To understand the molecular basis for the increase in cell proliferation in tadpole brain during early prometamorphosis we looked for cell cycle genes that were upregulated during metamorphosis, and cell cycle genes that were induced by T$_3$ treatment of NF54 tadpoles. Five *cyclin-dependent kinase* genes (*cdk14*, *cdk15*, *cdk17*, *cdk5r2*, *cdkl5*) were upregulated during metamorphosis (four had TR peaks), as were five other cell cycle genes (*cdc16*, *cdc27*, *cdkn1a*, *dedicator of cytokinesis 3—dock3*, and *regulator of cell cycle—rgcc*; only *dock3* had TR peaks). Of these, only *cdk14* and *rgcc* were induced by exogenous T$_3$ in NF54 tadpoles.

**Neural cell differentiation, cell structure and signaling:** Consistent with cell cycle exit and terminal differentiation, we saw downregulation of genes encoding proteins expressed in neural stem/progenitor cells [24]. These included *vimentin*, *nestin*, *notch 1 receptor* and 10 *hes* genes (homologs of Drosophila *hairy* and *Enhancer of split*, which encode basic helix-loop-helix transcriptional repressors) (Fig 5A and S11 Table). Only *hes4* had TR peaks.

Conversely, genes involved with neuronal differentiation, cell structure and signaling were upregulated during metamorphosis. These included *neuronal differentiation 6* (*neurod6*; has a TR peak), *bone morphogenetic protein 1* and *4*, *fibroblast growth factor 9* and *12*, *neurotrophic receptor tyrosine kinase 2* (has a TR peak) and several genes involved with synapse formation and function such as two *synaptosome associated protein* (*snap25* and *snap91*) and five *synaptotagmin* genes (two had TR peaks) (S11 Table). The RNA-seq expression profiles of 12 genes in this category are shown in Fig 5B. On the other hand, some genes involved with neural cell differentiation (and also cell proliferation) were downregulated such as *bmp receptor type 1B*, *fgf 3* and *11*, *fgf binding protein 3*, *fgf receptor 2*, *neurogenin 1* and *sonic hedgehog*.

There were many genes that encode proteins involved with signaling in the nervous system that were upregulated during metamorphosis. These included 11 *glutamate receptor* subunits (seven with TR peaks), 21 potassium channel-related proteins (seven with TR peaks), six calcium channel (two with TR peaks), *calmodulin 1* and *2* (both with TR peaks), four *calcium/calmodulin dependent protein kinase* (two with TR peaks), four sodium channel (one with TR peaks), four *transient receptor potential cation channel* (two with TR peaks) and both *P2X* and *P2Y purinergic receptor* (*p2ry1* has TR peaks) genes, among others (Fig 5B). Many genes

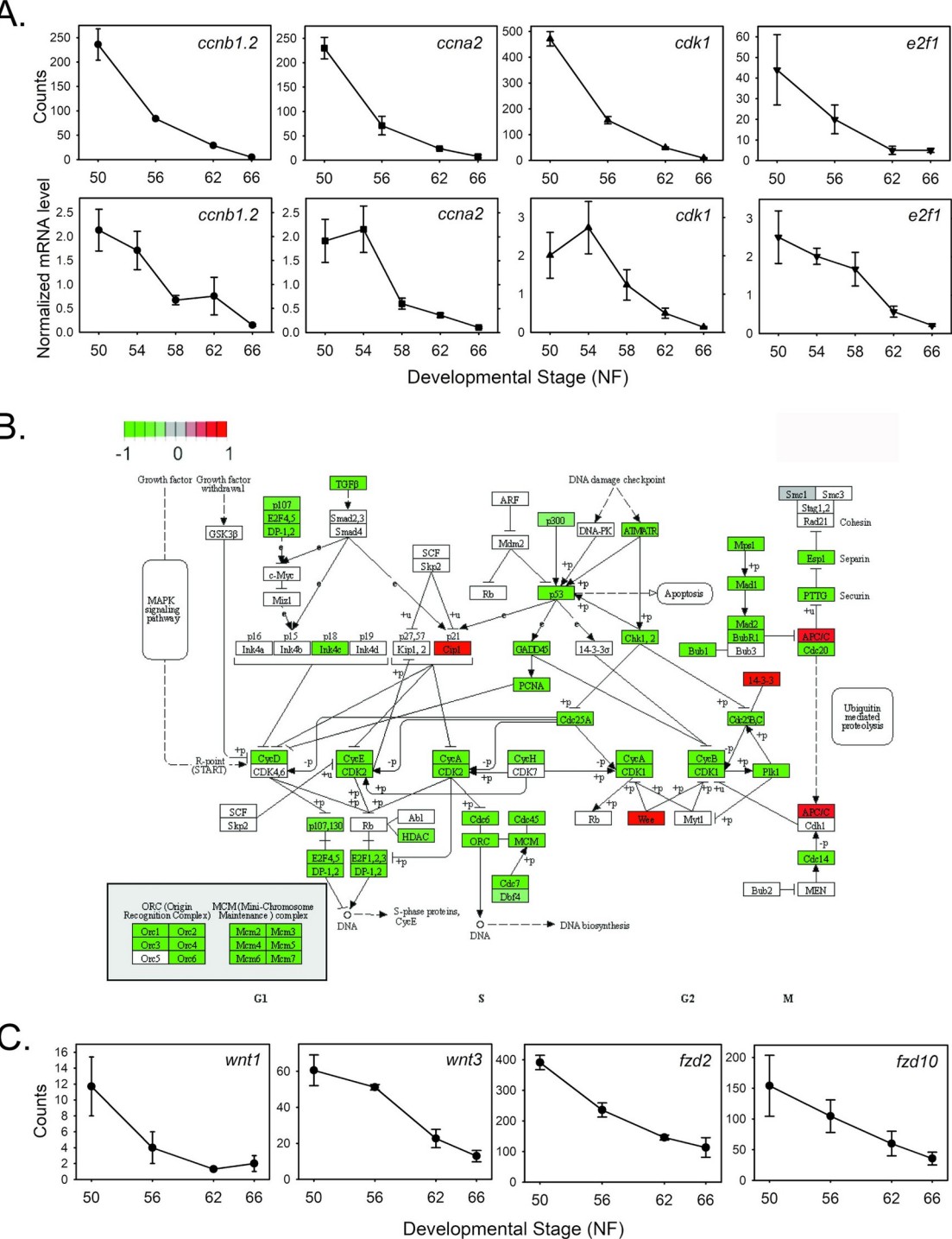

**Fig 4. Cell cycle control genes and genes encoding components of the Wnt/b-catenin signaling pathway are downregulated during metamorphosis. A.** Shown are the mean+SEM (n = 3/NF developmental stage) of RNA-seq count data (top) and RTqPCR data (bottom; n = 5/NF developmental stage) for four cell cycle control genes: *Cyclin b1—ccnb1.2*; *cyclin a2—ccna2*; *cyclin-dependent kinase 1—cdk1*; *E2F transcription factor—e2f1*. **B.** KEGG pathway analysis of RNA-seq data for gene expression changes during metamorphosis in *X. tropicalis* tadpole brain. Shown is the cell cycle control pathway. **C.** Shown are the mean+SEM (n = 3/ NF developmental stage) of RNA-seq count data for 4 genes: *Wnt family member 1—wnt1*; *Wnt family member 3—wnt3*; *frizzled class receptor 2—fzd2*; *frizzled class receptor 10—fzd10*.

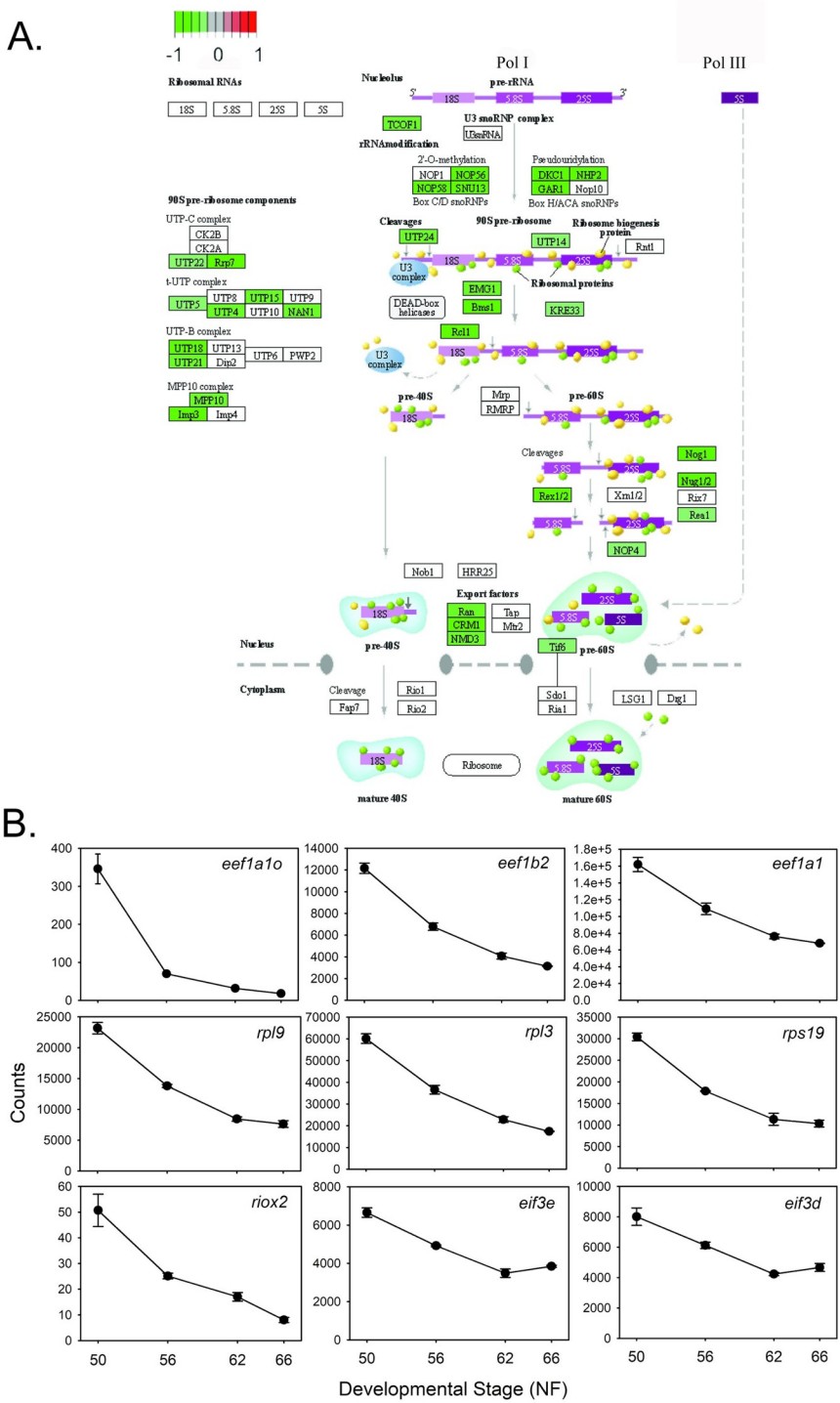

**Fig 5. Genes encoding stem/progenitor cell markers are downregulated, while genes encoding neural differentiation-related proteins are upregulated during metamorphosis. A.** Shown are the mean+SEM (n = 3/NF developmental stage) of RNA-seq count data for 6 stem/progenitor cell marker genes: *Vimentin—vim*; *nestin—nes*; *notch 1 receptor—notch1*; *hairy and enhancer of split 5 gene 4 –hes5.4*; *hairy and enhancer of split 5 gene 3—hes5.3*; *hairy and enhancer of split 6 gene 2—hes6.2*. **B.** Shown are the mean+SEM (n = 3/NF developmental stage) of RNA-seq count data for 12 neural differentiation-related genes: *Neuronal differentiation 6—neurod6*; *bone morphogenetic protein 1—bmp1*; *cholinergic receptor muscarinic 2—chrm2*; *transient receptor potential cation channel, subfamily M, member 8—trpm8*; *sodium channel, voltage gated, type V alpha subunit—scn5a*; *monoamine oxidase A—maoa*; *glutamate receptor, ionotropic, N-methyl-D-aspartate 3B - grin3b*; *calcium/calmodulin dependent protein kinase ID— camk1d*; *calmodulin 1—calm1*; *purinergic receptor P2Y, G-protein coupled, 1—p2ry1*; *purinergic receptor P2X, ligand gated ion channel, 5—p2rx5*; *myelin basic protein—mbp*.

encoding signaling molecules in the nervous system or their receptors were upregulated, such as serotonin, dopamine, acetylcholine, prostaglandins and gamma-aminobutyric acid.

**Protein synthesis:** The KEGG ribosome biogenesis pathway showed widespread downregulation during metamorphosis (Fig 6A) We identified 139 genes that encode ribosomal proteins and protein translation factors that were downregulated, indicating a large decline in protein synthesis during metamorphosis. These included 22 eukaryotic translation initiation factor (*eif*), 6 eukaryotic translation elongation factor (*eef*), and 76 ribosomal protein (*rp*) genes, among others. Shown in Fig 6B are the RNA-seq expression profiles for nine of these genes. Over half of the genes had TR peaks: 15 *eif*, three *eef* and 59 *rp* genes; this suggests a significant and possibly a direct role for TH-TR in down-regulating protein synthesis during metamorphosis.

**Neuroendocrine function:** Genes that encode neuroendocrine-related proteins were upregulated during metamorphosis, reflecting the maturation of these brain centers and the important role of neuroendocrine regulation for metamorphosis. We selected 57 upregulated genes in this category based on their known functions in hormone signaling; the RNA-seq expression profiles for 12 of these genes are shown in Fig 7. Of the 57 selected genes, 12 had TR peaks (S11 Table). The most strongly upregulated gene was *monodeiodinase type 3* (*dio3*; log2FoldChange = 3.275 from NF50 to NF62) (see Fig 2A); *dio3* did not have TR peaks within the set parameters. This was followed by *arginine vasotocin* (*avt*; log2FoldChange = 2.328), then inhibin subunit beta B (*inhbb*; log2FoldChange = 2.2); neither of these genes had TR peaks. The gene encoding the mineralocorticoid receptor (*nr3c2*), a known TH response gene in tadpoles [47], was upregulated (log2FoldChange = 1.57) and it had two TR peaks.

Several genes that are essential for reproduction were strongly upregulated during metamorphosis. These included the gene that encodes the neuropeptide *gonadotropin-releasing hormone 1* (*gnrh1*) and its receptor *gnrh receptor 2* (*gnrhr2*); genes encoding proteins involved with sex steroid metabolism and action like *steroidogenic acute regulator protein* (*star*), two *hydroxysteroid (17-beta) dehydrogenase* genes important for the generation of androgens (*hsd17b4* and *hsd17b6*), *estrogen receptor* 1 and 2 (*esr1* and *esr2*) and *progesterone receptor* (*pgr*; only *star* has a TR peak); and several other reproduction-related genes (e.g., two activin A receptors—*acvr1* and *acvr1C*, *follistatin like 3*—*fstl3*, *inhbb*, *prolactin*—*prl.1*, among others; S11 Table).

Genes that transduce the actions of neurohormones on the thyroid and interrenal axes (among other actions) like the *type I corticotropin-releasing hormone receptor* (*crhr1.2*) and *thyrotropin-releasing hormone receptor 3* were upregulated during metamorphosis. The mRNAs for genes

involved with energy balance and the regulation of food intake also increased, such as *melanin concentrating hormone*, *melanin-concentrating hormone receptor 1*, *cholecystokinin*, *adiponectin receptor 1*, *hypocretin (orexin) receptor 2*, *melanocortin receptor 4*, *neuropeptide y receptor Y2*, among others. Other neuropeptide, neuropeptide receptor and steroid metabolism genes were similarly upregulated during metamorphosis, and some had TR peaks (S11 Table). Many fewer neuroendocrine-related genes were downregulated in tadpole brain during metamorphosis (11 selected) and none of these had TR peaks.

## Discussion

Here we report the first comprehensive, genome-wide analysis of gene expression changes using RNA-seq in an amphibian tadpole tissue from the beginning to the end of spontaneous metamorphosis. We also identified TR binding sites across the tadpole neural cell genome using ChIP-seq, and we compared gene expression changes during metamorphosis with those

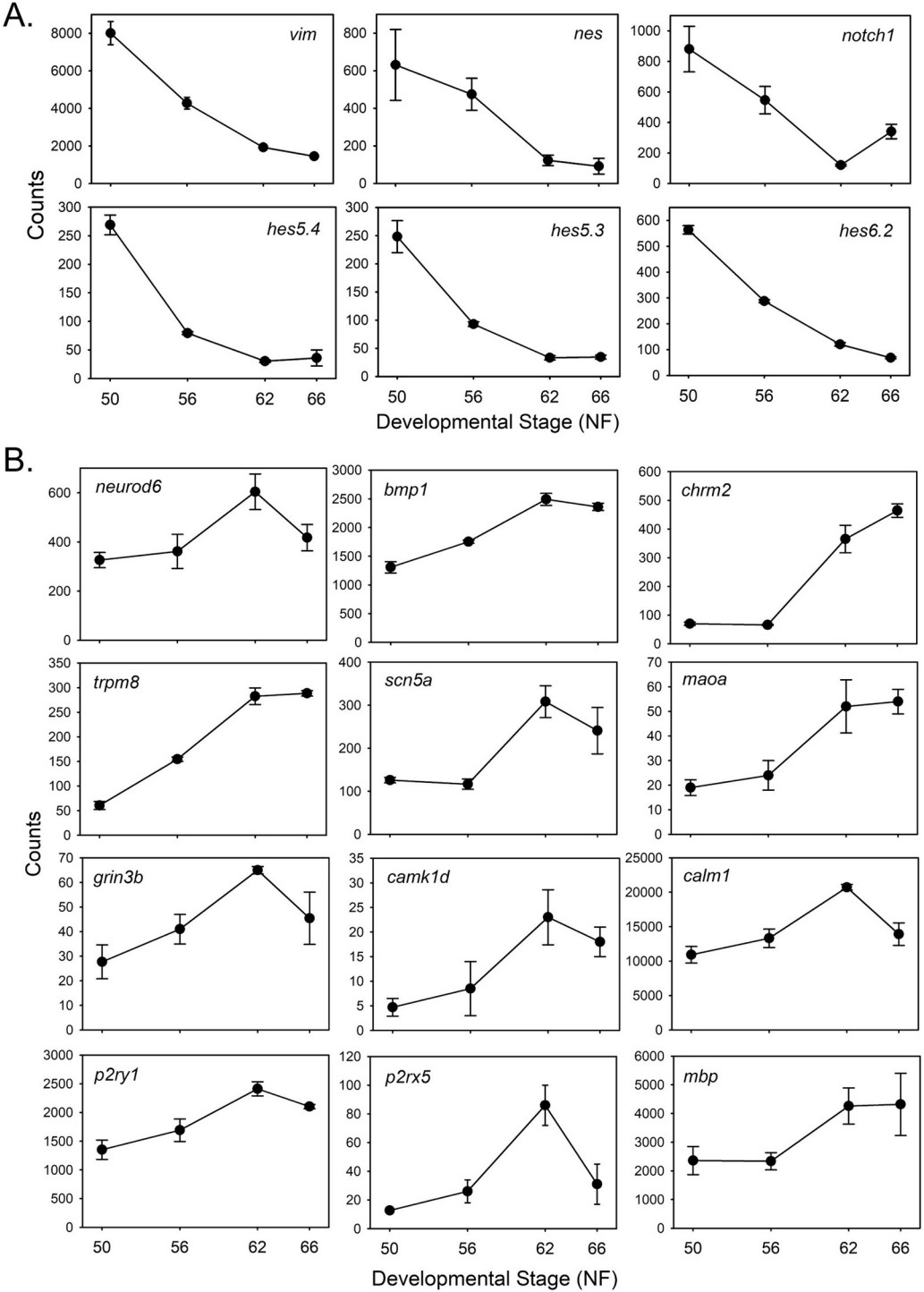

**Fig 6. Genes encoding proteins involved with ribosome biogenesis and protein synthesis are downregulated during metamorphosis. A.** KEGG pathway analysis of RNA-seq data for gene expression changes during metamorphosis in *X. tropicalis* tadpole brain (preoptic area/thalamus/hypothalamus). Shown is the ribosome biogenesis pathway. **B.** Shown are the mean+SEM (n = 3/NF developmental stage) of RNA-seq count data for 9 genes: *Eukaryotic translation elongation factor 1 alpha 1—eef1a1o*; *eukaryotic translation elongation factor 1 beta 2—eef1b2*; *eukaryotic translation elongation factor 1 alpha 1—eef1a1*; *ribosomal protein L9—rpl9*; *ribosomal protein L3—rpl3*; *ribosomal protein S19- rps19*; *ribosomal oxygenase 2- riox2*; *eukaryotic translation initiation factor 3 subunit E—eif3e*; *eukaryotic translation initiation factor 3 subunit D—eif3d*.

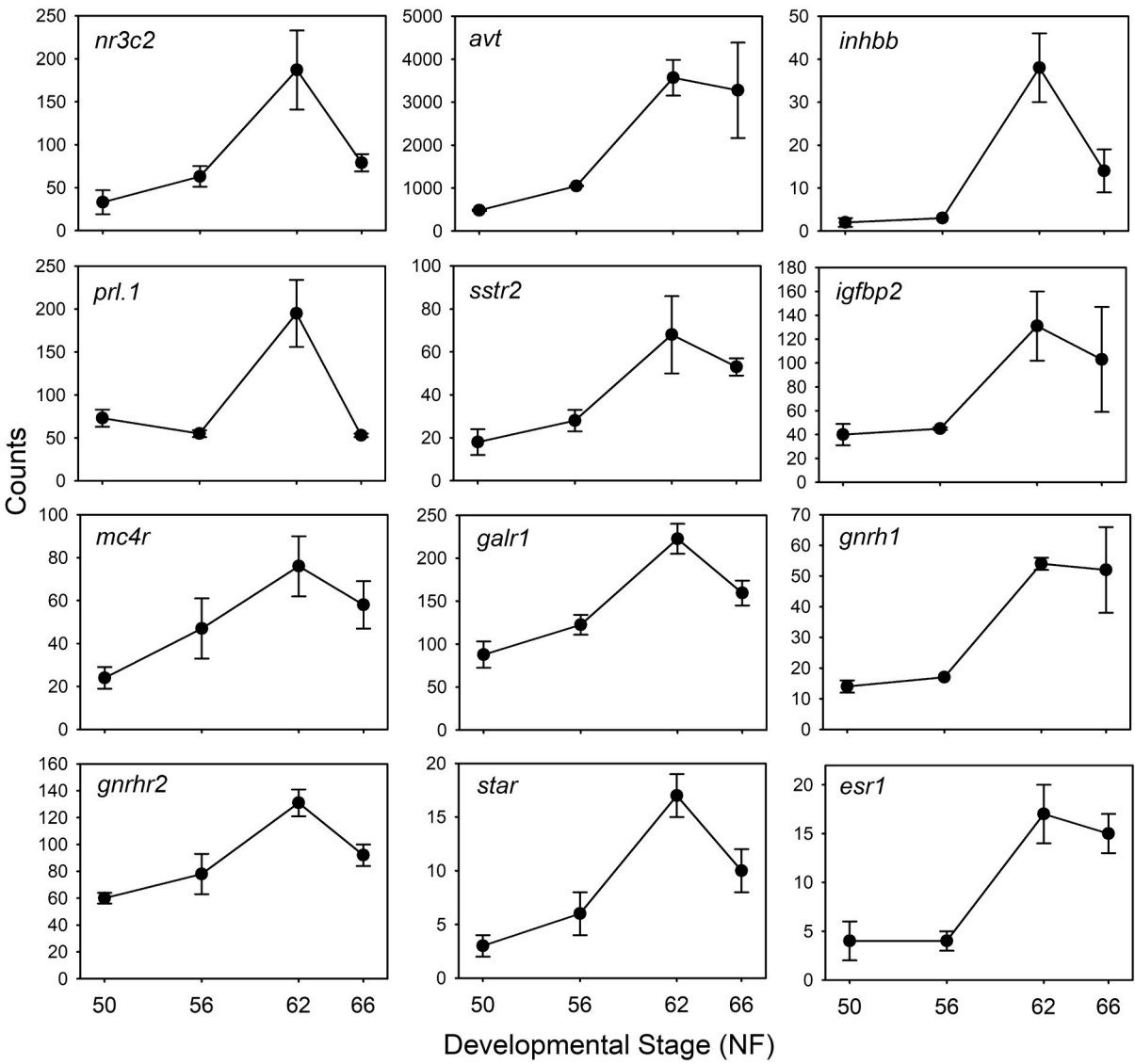

**Fig 7. Genes encoding neuroendocrine-related proteins are upregulated during metamorphosis.** Shown are the mean±SEM (n = 3/NF developmental stage) of RNA-seq count data for 12 genes: *Nuclear receptor subfamily 3 group C member 2* (mineralocorticoid receptor)— *nr3c2*; *arginine vasotocin—avt**; *inhibin subunit beta B—inhbb*; *prolactin, gene 1—prl.1*; *somatostatin receptor 2—sstr2*; *insulin like growth factor binding protein 2—igfbp2*; *melanocortin 4 receptor—mc4r*; *galanin receptor 1—galr1*; *gonadotropin releasing hormone 1—gnrh1*; *gonadotropin releasing hormone receptor 2—gnrhr2*; *steroidogenic acute regulatory protein—star*; *estrogen receptor 1—esr1*. *This gene is mislabeled in the genome database as *arginine vasopressin* (*avp*) which is a mammalian gene. The amphibian gene is *arginine vasotocin* (*avt*).

caused by treatment of premetamorphic tadpoles with $T_3$. We focused on the preoptic area/ thalamus/hypothalamus of *X. tropicalis* tadpole brain, which among other functions, houses the neuroendocrine centers that control pituitary hormone secretion necessary for metamorphosis, and whose maturation depends on the actions of TH [46,48,49]. The gene regulation changes that we identified underlie the spectrum of biochemical, physiological and morphological changes that occur in tadpole brain during metamorphosis. Taken together, our findings provide a foundation for understanding the molecular basis for metamorphosis of tadpole brain, but also raise questions about the physiological significance of the gene regulation changes induced by exogenous $T_3$ in premetamorphic tadpoles.

## Patterns of gene expression changes in tadpole brain during metamorphosis

The mRNA levels of approximately one fourth of the *X. tropicalis* protein coding genes changed their mRNA levels in tadpole brain during metamorphosis; about half of the genes were upregulated and half downregulated. Our findings, using an unbiased approach that spanned premetamorphosis to the completion of spontaneous metamorphosis, support earlier hypotheses that were based on analyses of a small number of individual genes and the timing of tissue transformations, that the most active period of gene regulation in tadpole brain is from the beginning of metamorphosis to metamorphic climax when circulating TH increases dramatically and reaches a peak, then declines [4,5,11,24,50].

## Identification of TR association in chromatin using TR ChIP-seq

There are limited data on TR association in chromatin across the genome of any species, and the number of peaks reported varies widely among studies owing to the different species (mouse, human and *X. tropicalis*) and tissues analyzed (liver cells, neuronal cells, intestine), differences in parameters used for peak calling (numbers of peaks ranging from $\sim$1100 to $\sim$60,000), using antiserums or monoclonal antibodies (i.e., ChIP) vs. affinity purification of tagged TR proteins, among other biological and technical reasons [43,51–55]. Nevertheless, there are some common findings among these and the current study that allow for the development of several testable hypotheses.

We asked two main questions in our TR ChIP-seq experiment: how many TR peaks are associated with annotated genes, and how many genes that change their mRNA levels during metamorphosis or after $T_3$ treatment have TR peaks? We found 6302 unique peaks in metamorphic climax stage tadpoles, and most of these peaks (81%) were associated with genes using the window of discovery of 10 kb upstream of TSSs and within gene bodies (some TR peaks were associated with more than one gene, and some genes had multiple TR peaks.) Approximately two thirds of the TR peaks were clustered +/- 1 kb from the TSSs of the associated genes, which is similar to results obtained in mammalian cells [56], suggesting that a majority of TREs are proximal to TH-regulated genes.

Of the TR peaks that were associated with genes, only 35% were associated with DEGs whose mRNA levels changed during metamorphosis; this was only 10% for TR peaks associated with genes whose mRNAs were modulated by exogenous $T_3$ in premetamorphic tadpole brain. Tanizaki and colleagues [43] used TR ChIP-seq on intestine of NF56 *X. tropicalis* tadpoles treated with or without $T_3$ for 18 h, and similar to our findings in tadpole brain, they found only 27% of TR peaks were associated with genes whose mRNA levels changed after $T_3$ treatment (they assigned peaks to genes if the peaks were 5 kb upstream of the TSS or within gene bodies). Thus, in tadpole intestine and brain, over two thirds of the TR peaks associated with genes were at genes whose mRNAs did not change during spontaneous metamorphosis or after $T_3$ treatment.

Similarly, studies conducted on mammalian cerebellum and liver-derived cells found that a majority of TR peaks were associated with genes whose mRNAs were unaffected by exogenous $T_3$ (Ayers et al., 2014; Chatonnet et al., 2013; Grøntved et al., 2015; Ramadoss et al., 2014). Taken together, these findings suggest that many genes with proximal TR peaks in ChIP-seq experiments have TREs that do not function to transduce TH action, at least under the conditions of the experiment; or, many TREs do not regulate the genes with which they are associated based on the criteria that were used. These putative TR binding sites could function under different developmental or physiological states (i.e., the chromatin in the stage/condition under study might be compartmentalized by insulators) [57–59], or in other tissues not

yet studied, or they may regulate genes other than those that were assigned using the set criteria, perhaps via long-range chromosome interactions [60,61]. Each of these possibilities can be tested experimentally.

Most DEGs in tadpole brain during spontaneous metamorphosis or after $T_3$ treatment did not have TR peaks (75.6% and 90%, respectively; see S6 Table). Similar results were obtained by Tanizaki and colleagues [43] with tadpole intestine, and Chatonnet and colleagues [62] with mouse cerebellum-derived C17.2 cells engineered to express TRs. There are several possible explanations for these findings. First, many of the DEGs may not be directly regulated by the TR complex, but instead are secondary or tertiary TH response genes [56]. Second, some genes that change their mRNA levels during metamorphosis may depend on other factors unrelated to TH action. Third, some DEGs may be regulated by TREs located outside of the window of discovery that we used here; that is, they regulate the DEG via long range chromosome interactions [60,61]; 19% of TR peaks in tadpole brain were not associated with genes using the criteria applied here. Fourth, nuclear receptors show dynamic interactions with chromatin that cannot be captured at a single time point, which may have caused us to miss TR association at some regulated genes (e.g., for evidence for dynamic TR association in chromatin see [53]). Lastly, some genes may lack TREs, but be regulated by TH signaling via nongenomic pathways [45].

We found approximately equal numbers of up- and down-regulated genes with TR peaks in tadpole brain during metamorphosis and in response to exogenous $T_3$, which is similar to the findings of Tanizaki and colleagues [43] in tadpole intestine (they reported 36% up- and 26% down-regulated genes with TR peaks). These findings in tadpoles contrast with mouse liver where only a small number of the total TR peaks were found at $T_3$-repressed genes [52,53,56].

## Cellular pathways regulated during metamorphosis

We found changes in the mRNA levels of genes representing several cellular pathways and structural components that underlie the cell and tissue maturational processes occurring in tadpole brain during metamorphosis. The gene expression changes show that during metamorphosis of tadpole brain there is a shift from building neural structures (i.e., mitosis, protein synthesis) early in the metamorphic process, to the differentiation and maturation of neural cells and neural signaling pathways characteristic of the adult frog brain (i.e., cell differentiation, synaptogenesis, cell-cell signaling, neurosecretion, etc.) Clustering analysis revealed five major patterns of gene regulation, two clusters of downregulation, two of upregulation, and one smaller cluster with genes that were first upregulated then downregulated.

The two clusters of downregulated genes include proteins involved with cell cycle, protein synthesis, DNA replication and DNA repair. Previous studies showed that during early prometamorphosis there is an increase in cell proliferation in subventricular zones of tadpole brain that depends on endogenous TH acting via TRa [21,46,63]. After this developmental period (after NF56-57), there is a progressive decline in mitosis through the end of metamorphosis; tadpoles become refractory to exogenous $T_3$ action on cell proliferation after NF54-55 [46]. We found that genes that encode proteins involved with cell cycle and cell division were among the most strongly downregulated after NF56-57. These included myc protooncogene, aurora kinase, cyclin, cell division cycle, cyclin-dependent kinase, E2F transcription factor and other genes. Some of these cell cycle genes could be directly regulated by the TH-TR complex, as evidenced by our TR ChIP-seq experiment. This dramatic decline in cell cycle gene expression after early prometamorphosis provides a molecular basis for earlier experiments that showed a decline in cell proliferation in tadpole brain as metamorphosis progressed [46] (see

also [64]). Similar findings from RNA-seq experiments showing decreases in cell cycle gene mRNA levels were recently reported for *X. tropicalis* tadpole intestine [43] and *X. laevis* tadpole midbrain [24].

We also saw downregulation during metamorphosis of genes encoding multiple components of the Wnt/b-catenin signaling pathway, which is linked to cell cycle control via cyclin-dependent kinases [65,66], and has been implicated in TH-stimulated cell proliferation in mouse hepatocytes (Francavilla et al., 1994; Pibiri et al., 2001; Hones et al., 2022). A time and dose-dependent repression of Wnt/b-catenin related genes by $T_3$ was reported in rat pituitary-derived GC cells [67], suggesting that this pathway is a key target of TH signaling. This pathway may also be negatively regulated by nongenomic TH signaling [68]. In addition to downregulation of cell cycle and Wnt/b-catenin signaling pathway genes, a cyclin-dependent kinase inhibitor 1A (*cdki1a*) was upregulated in tadpole brain during metamorphosis. Thyroid hormone-dependent upregulation of a cyclin-dependent kinase inhibitor ($p27^{Kip1}$) promotes cell cycle exit in mouse neuroblastoma cells engineered to express TRb1 [69]. Thus, the TH-dependent downregulation of cyclins and other cell-cycle genes, and the upregulation of inhibitors of cyclin-dependent kinases together may promote cell cycle exit in the developing tadpole nervous system.

What signaling pathways are responsible for the TH-dependent increase in cell proliferation in tadpole brain during early prometamorphosis? To address this question, we looked for cell cycle genes that were upregulated during metamorphosis, and cell cycle genes that were induced by $T_3$ treatment of premetamorphic tadpoles. Five genes encoding cyclin-dependent kinases and five other cell cycle-related genes were upregulated during early prometamorphosis, and of these, the mRNA levels of two, *cyclin-dependent kinase 14* (*cdk14*) and *regulator of cell cycle* (*rgcc/rgc-32*) were induced by exogenous $T_3$ in premetamorphic tadpole brain. The *cdk14* gene encodes a serine/threonine-protein kinase involved with eukaryotic cell cycle control whose activity is controlled by an associated cyclin (cyclin D3), and it is the only cyclin-dependent kinase known to activate the Wnt/b-catenin signaling pathway [66]. The *rgcc* gene encodes a protein that regulates cell cycle and transduces the actions of transforming growth factor b [70]. We hypothesize that these two proteins play key roles in the actions of TH on cell proliferation in early prometamorphic tadpole brain.

We also saw broad downregulation of genes encoding proteins involved with protein synthesis during metamorphosis. These included ribosomal proteins and protein translation factors, many of which had TR peaks, suggesting that they may be directly regulated by TH-TR. Most striking were the eukaryotic initiation factor and ribosomal protein genes, of which 71% (15 of 21 genes) and 78% (59 of 76 genes), respectively, had TR peaks. These findings support that protein synthesis is strongly downregulated during metamorphosis, and could be mediated in large part by the direct, genomic actions of TH. The decline in protein synthesis could reflect, in part, the negative energy balance during metamorphosis, when tadpoles cease eating as the gut undergoes dramatic remodeling to accommodate the shift from an herbivorous/omnivorous to a carnivorous feeding ecology [71].

Reflecting a shift from cell proliferation to cell differentiation and maturation, many genes that are expressed in neural stem/progenitor cells were downregulated during metamorphosis like *vimentin* [72], *nestin* [73], *notch 1 receptor* and 11 *hes* genes (hairy and enhancer of split, encoding basic helix-loop-helix transcriptional repressors; some Hes proteins regulate the notch signaling pathway) [74]. By contrast, genes encoding proteins involved with neural cell differentiation (e.g., *neurod6*; discussed below), and proteins expressed in mature neurons such as ion channels, neurotransmitter receptors, enzymes involved with intracellular signaling, etc., and myelination-related glial markers like *myelin basic protein* were upregulated during metamorphosis. Ta and colleagues [24] recently reported similar results in *X. laevis*

midbrain. These findings, paired with earlier histochemical and biochemical analyses, clearly demonstrate the developmental transition occurring in tadpole brain under the control of TH: exit from the cell cycle followed by cell differentiation, and maturation of neural cells.

Two neural differentiation (*neurod*) genes were regulated during metamorphosis. There are four members of this protein family in vertebrates (Neurod1, Neurod2, Neurod4 and Neurod6) and they play pivotal roles in neurogenesis and neuronal progenitor cell differentiation and specification [75]. We found that the mRNA levels for *neurod1* and *neurod2* did not change during tadpole metamorphosis (although *neurod2* mRNA was induced by exogenous $T_3$). However, the *neurod6* mRNA level increased during metamorphosis and in response to exogenous $T_3$, and it had TR peaks, supporting that it is directly induced by the TH-TR complex. In mammals, Neurod6 has important functions in neuronal differentiation and axon development [75]. We hypothesize that the direct regulation of *neurod6* by TH in *X. tropicalis* is a key factor in regulating neuronal differentiation in the tadpole nervous system. By contrast, *neurod4* mRNA declined during metamorphosis, but was not regulated by exogenous $T_3$, nor did it have TR peaks. In mammals, Neurod4 functions during early stages of neurogenesis [75], and if this is conserved in frogs, it would be consistent with its downregulation during metamorphosis.

Many genes encoding proteins involved with neuroendocrine signaling, which play central roles in controlling the timing of metamorphosis, among other critical physiological functions, and several genes that encode proteins that function in TH signaling were upregulated in tadpole brain during metamorphosis. Notably, three genes that were among the five most strongly upregulated during metamorphosis included *arginine vasotocin* (*avt*), which encodes a precursor that gives rise to a nonapeptide involved with osmoregulation and pituitary hormone secretion, monodeiodinase type 3 (*dio3*) which functions to inactivate TH [4], and TH induced bZip protein (*thibz*), a basic leucine-zipper transcription factor that is one of the most strongly upregulated TH-target genes in tadpole tissues [37]. Several other genes that encode neuropeptides and neuropeptide receptors, nuclear receptors and steroid metabolizing enzymes that function in reproduction, food intake and energy balance, among other functions, were upregulated, reflecting the maturation of these systems during metamorphosis [71,76,77].

The shift in the actions of TH on the balance between cell proliferation and cell cycle exit/ cell differentiation occurs in multiple species and in different tissues [45]. We hypothesize that this shift in tadpoles is driven by the dramatic increase in *thrb* expression that occurs during prometamorphosis [78]. Prior to this period, and shortly after hatching, *thra* is expressed at a relatively high and constant level in tadpole tissues, but *thrb* mRNA remains low or nondetectable until prometamorphosis [31]. During prometamorphosis there is a large increase in *thrb* expression driven by the rise in circulating TH concentration (autoinduction) [36,78,79]. The TRa subtype mediates the actions of TH on cell proliferation in tadpole brain [21,46] and in mammalian neural cells [80–82]. By contrast, liganded TRb promotes cell cycle exit and cell differentiation [46,69,83–86]. Thus, rising *thrb* gene expression and plasma TH titers together may induce the shift from cell proliferation to cell cycle exit and differentiation.

## Comparison of the developmental and $T_3$-regulated gene expression programs

The experimental paradigm in which premetamorphic tadpoles are exposed to thyroid gland homogenate or synthetic TH by feeding, injection or addition to the aquarium water has been used by investigators for over a century, since the time that Gudernatsch [87] fed thyroid glands harvested from livestock to tadpoles and saw accelerated metamorphosis. The tissue hormone concentration achieved, and the effects of the exogenous hormone were thought to

recapitulate the changes that occur at metamorphic climax under the influence of endogenous TH. However, it should be noted that the dosage of $T_3$ that is used in many experiments (5 nM), which was chosen to recapitulate the endogenous circulating hormone concentration at metamorphic climax (estimated to be $\sim 8$ nM) [88], produces tissue content of $T_3$ that is 3–4 times the nominal concentration of the hormone in the aquarium water, suggesting that tadpoles take up and concentrate the hormone from their environment [89]. Also, this dosage of $T_3$ typically causes asynchronous tissue transformations that leads to death of the animals within a week of exposure to the hormone.

Gene expression screens that were designed to investigate $T_3$-induced tissue transformations in amphibian tadpoles were first conducted in the 1990s using hybridization-based methods like subtractive hybridization [6–11] and DNA microarrays [12–20]. Each of these studies examined gene regulation responses in premetamorphic tadpole tissues following treatment with $T_3$ for different times, and the findings were hypothesized to correspond to the endogenous gene regulation programs during metamorphosis. However, to our knowledge, there has not been a direct test of this hypothesis; i.e., that the gene regulation changes induced by exogenous $T_3$ correspond to those that occur during spontaneous metamorphosis. In each of these studies a small number of genes were validated using targeted approaches like Northern blotting, semi-quantitative and quantitative PCR; no studies indicated what percentage of the genes did not validate.

To address the question of whether and to what extent the $T_3$-induced gene regulation changes reflect normal developmental processes, we compared changes in mRNA levels in tadpole brain during spontaneous metamorphosis [25] to those occurring after 16 h of $T_3$ treatment of premetamorphic tadpoles [21]. For both RNA-seq studies we used tadpoles from the same laboratory breeding colony, and the same RNA extraction methods and source of reagents, sequencing chemistry and platform, and bioinformatics analyses. We recognize that technical variation between our two RNA-seq experiments could explain some of the differences that we noted between the developmental and $T_3$-induced RNA-seq datasets. However, the magnitude of the differences that we discovered, and the low concordance in gene expression responses (i.e., the direction of change in mRNA levels after $T_3$ treatment vs. spontaneous metamorphosis), suggests that the $T_3$ treatment paradigm could be prone to artifact, and therefore should be recognized and tested in future studies.

In our analysis we found that two thirds of the genes that changed their mRNA levels during spontaneous metamorphosis were not modulated by exogenous $T_3$ in premetamorphic tadpole brain at the time point that we analyzed. There are several possible technical or biological explanations for our findings. First, since we analyzed a single time point we may have missed some genes whose mRNA levels were modulated by $T_3$ at earlier or later time points. We designed the experiment based on previous kinetic analyses of individual gene regulation responses to $T_3$ in premetamorphic tadpoles to capture the immediate-early and delayed immediate-early transcriptional responses to the hormone, which comprise many direct TH response genes (likely also some secondary response genes; i.e., indirect TH response genes). Second, perhaps many of the genes that change during metamorphosis are secondary or tertiary TH response genes (i.e., indirect response genes), or are regulated by other factors independent of TH. Third, it is possible that some genes that change their expression during spontaneous metamorphosis are normally regulated by TH, but do not respond to the hormone in premetamorphic tadpoles owing to the presence of a heterochromatic state at the gene's TRE or other regulatory sites, and/or a lack of TR occupancy at the gene's TRE(s). These genes may require TH action at other loci to produce proteins that modify chromatin, and consequently promote transcriptional responses at some $T_3$ target genes [90]. For example, DNA methylation plays a pivotal role in modulating chromatin structure and gene

transcription, and we recently showed that the tadpole neural cell genome undergoes dynamic DNA methylation changes, with DNA demethylation predominating at metamorphic climax [25]. Such changes may allow genes that are quiescent in premetamorphic tadpole brain to become responsive to TH. Concordantly, the mRNAs of genes that encode enzymes that catalyze histone modifications are upregulated in tadpole tissues during metamorphosis, and some are induced precociously in premetamorphic tadpoles by exogenous $T_3$ [91,92].

On the other hand, of the genes that were regulated by $T_3$ in premetamorphic tadpole brain, the mRNA levels of approximately half of these did not change during metamorphosis. Furthermore, of the genes that overlapped between the developmental and $T_3$-induced datasets, the concordance (i.e., the direction of change in gene expression, up- or downregulated) was as low as 50% depending on the developmental interval analyzed. That is, many genes whose mRNAs changed during spontaneous metamorphosis and were also modulated by exogenous $T_3$ in premetamorphic tadpoles changed in the opposite direction. The percentage of genes whose direction of change was in concordance was greatest during late prometamorphosis (NF56-62; 81%) when plasma TH titers rise to peak at metamorphic climax. Taken together, these findings suggest that some gene regulation responses to $T_3$ in premetamorphic tadpoles could represent pharmacological responses to the exogenous hormone. Our findings suggest that pathway analysis of data from experiments conducted with exogenous $T_3$ may not be accurate representations of normal developmental processes.

## Supporting information

**S1 Fig. Region of the tadpole brain dissected for RNA extraction and analysis by RNA-seq at four stages of metamorphosis, or after $T_3$ treatment of premetamorphic tadpoles.** The coronal anatomical diagrams of *Xenopus* brain are from Tuinhof and colleagues [93] with modifications by Yao and colleagues [94]. Dotted lines demarcate the region of the tadpole brain that was dissected. Abbreviations: Apl, Amygdala pars lateralis; BNST, bed nucleus of the stria terminalis; C, central thalamic nucleus; CeA, central amygdala; dp, dorsal pallium; LA, lateral amygdala; LH, Lateral hypothalamus lp, lateral pallium; Lpv, lateral thalamic nucleus, pars posteroventralis; MeA, medial amygdala; mp, medial pallium; NPv, nucleus of the paraventricular organ; nII, cranial nerve II; OB, olfactory bulb; OT, optic tectum; P, posterior thalamic nucleus; POa, preoptic area; Tel, telencephalon; VH, ventral hypothalamic nucleus; VM, ventromedial thalamic nucleus.
(TIF)

**S2 Fig. Validation of TR ChIP-sequencing conducted on brain chromatin from X. tropicalis tadpoles at metamorphic climax using targeted ChIPqPCR. A.** Genome browser traces showing the location of TR peaks at three uncharacterized loci. The genomic coordinates are given in the table below the image (1 –XLOC_019343.1; 2 –XLOC_036006.1; 3 – XLOC_032902.1). Bars below the traces indicate the predicted exons and lines the predicted introns. **B.** Targeted ChIPqPCR assays for TR at the genomic locations shown in panel A using chromatin isolated from brains of tadpoles at metamorphic climax (NF stage 62). The oligonucleotide primers used are shown in the table below the graph. Previously characterized TREs at the *tet2* and *tet3* loci were included as controls [32]. Chromatin extracts were precipitated using anti-TR serum or normal rabbit serum (NRS) as a control. Bars represent the mean$\pm$SEM of the ChIP signal expressed as a percentage of the input (n-4/treatment). Asterisks indicate statistically significant differences between the anti-TR serum and NRS ($p<0.05$; Student's t-test).
(TIF)

**S3 Fig. Thyroid hormone receptor (TR) associates in chromatin at the *Krüppel-like factor 9 (klf9)* locus in tadpole brain at metamorphic climax.** Shown are Integrative Genome Viewer (IGV) genome browser tracks for TR ChIP-seq reads mapped to the *Xenopus tropicalis* genome. We conducted a TR ChIP-seq experiment on chromatin isolated from the region of the preoptic area/thalamus/hypothalamus of metamorphic climax stage (NF stage 62) *X. tropicalis* tadpole brain. The input tracks are shown above the TR ChIP-seq tracks. Numbers in parentheses represent the scale for peak height. The gene structures are shown below the genome traces; lines and black filled bars represent introns and exons, respectively, and arrows indicate the direction 5' → 3'. In addition to the previously discovered TR association at the upstream *klf9* synergy module (KSM; which contains a DR+4 $T_3$ response element located ∼6 kb upstream of the transcription start site) [39] we observed TR ChIP-seq peaks 5' to the KSM, near the transcription start site (TSS) and within the gene. These additional sites of TR association may represent previously uncharacterized thyroid hormone response elements, or perhaps apparent TR association at this region caused by chromosomal looping [39].
(TIF)

**S1 Table. RNA sequencing data.**
(XLSX)

**S2 Table. Pair-wise comparisons of regulated genes at different developmental intervals.**
(XLSX)

**S3 Table. Comparisons of numbers and patterns of gene expression changes during metamorphosis and after T3 treatment of premetamorphic tadpoles.**
(XLSX)

**S4 Table. All TR ChIP-seq peaks with coordinates.**
(XLSX)

**S5 Table. Accounting of all TR peaks; peak + gene association; total TR peaks and DEGs for all of metamorphosis, prometamorphosis and climax (NF50 to NF62), and T3 treatment of premetamorphic tadpoles.**
(XLSX)

**S6 Table. Comparison of DEGs with and without TR peaks by developmental stage interval.**
(XLSX)

**S7 Table. Direct repeat +4 thyroid hormone response element motifs are enriched in the TR ChIP-seq peaks in tadpole brain.** Using HOMER software we found that the direct repeat with 4 base spacer (DR4) THRa NR and THRb NR motifs were enriched in the DNA sequences within the TR ChIP-seq peaks. Note that the HOMER software did not show the first nucleotide present in the first 6-base half-site for the THRa and THRb motifs.
(DOCX)

**S8 Table. gProfiler analysis of RNA-seq metamorphosis data by gene regulation cluster.**
(XLSX)

**S9 Table. gProfiler analysis of RNA-seq metamorphosis data by stage comparison.**
(XLSX)

**S10 Table. gProfiler analysis of RNA-seq T3 treated data.**
(XLSX)

**S11 Table. Cellular processes regulated during metamorphosis organized by category.**
(XLSX)

**S1 File.**
(DS_STORE)

## Acknowledgments

Dr. Yun-Bo Shi kindly provided the anti-TR serum.

## Author Contributions

**Conceptualization:** Samhitha Raj, Christopher J. Sifuentes, Yasuhiro Kyono, Robert J. Denver.

**Data curation:** Samhitha Raj, Christopher J. Sifuentes, Yasuhiro Kyono, Robert J. Denver.

**Formal analysis:** Samhitha Raj, Christopher J. Sifuentes, Yasuhiro Kyono, Robert J. Denver.

**Funding acquisition:** Robert J. Denver.

**Investigation:** Samhitha Raj, Christopher J. Sifuentes, Yasuhiro Kyono.

**Project administration:** Robert J. Denver.

**Software:** Christopher J. Sifuentes.

**Supervision:** Robert J. Denver.

**Validation:** Samhitha Raj, Christopher J. Sifuentes.

**Visualization:** Christopher J. Sifuentes, Robert J. Denver.

**Writing – original draft:** Samhitha Raj, Christopher J. Sifuentes, Robert J. Denver.

**Writing – review & editing:** Samhitha Raj, Christopher J. Sifuentes, Yasuhiro Kyono, Robert J. Denver.

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
