## [Decision Letter · Decision Letter 0]

22 May 2023

PONE-D-23-06337Metamorphic gene regulation programs in Xenopus tropicalis tadpole brainPLOS ONE

Dear Dr. Denver,

Thank you for submitting your manuscript to PLOS ONE. After careful consideration, we feel that it has merit but does not fully meet PLOS ONE’s publication criteria as it currently stands. Therefore, we invite you to submit a revised version of the manuscript that addresses the points raised during the review process.

We look forward to receiving your revised manuscript.

Kind regards,

Laurent Coen, Ph.D.

Academic Editor

PLOS ONE

“Dr. Yun-Bo Shi kindly provided the anti-TR serum. This research was supported by NSF grant IOS 1456115 to RJD.”

“This research was supported by National Science Foundation grant IOS 1456115 to RJD. https://www.nsf.gov/ The funders had no role in study design, data collection and analysis, decision to publish, or preparation of the manuscript.”

Additional Editor Comments:

Dear Robert J. Denvers,

Thank you for submitting your manuscript to PLOS ONE. After careful consideration, I feel that it is suitable for publication with some clarification and proofreading. Therefore, my decision is "Minor Revision."

First, I would like to apologize to the authors for the length of time the review process has taken and thank them for their patience.

I also want to give a brief explanation for the long delay:

After receiving the reports from two reviewers, I asked for the opinion of a third reviewer, as the first reviews resulted in a completely opposite opinions from the 2 reviewers, one being very positive and the other calling for rejection of the article. In this situation, I sought another independent opinion so that I could make my decision in full transparency with regard to these evaluations.

In their comments, two of the reviewers are very positive and agree that the work is of good quality and of interest. They underline the relevance of the data presented and feel that this is an important work.

The question of whether the events that occur during natural metamorphosis and induced metamorphosis are comparable is clearly an important issue that deserves to be addressed and debated.

Both reviewers stress this point and I would like to join them in saying that this central question in the article is a very interesting one and deserves to be opened to the scientific community.

However, if the reviewers are globally enthusiastic, the work raises some questions and comments as pointed out in some of the comments of reviewers 1 and 3, which deserves to be reconsidered and improved. There is therefore a need to revise the manuscript substantially before publication.

In conclusion, I ask the authors to give the most complete answer to the reviewers, trying as much as possible to provide arguments to answer their different questions, especially regarding the comments of reviewer 1. In addition, there is some proofreadings that must be corrected.

Reviewers' comments:

Reviewer's Responses to Questions

**Comments to the Author**

1. Is the manuscript technically sound, and do the data support the conclusions?

Reviewer #1: No

Reviewer #2: Yes

Reviewer #3: Yes

2. Has the statistical analysis been performed appropriately and rigorously? 

Reviewer #1: Yes

Reviewer #2: Yes

Reviewer #3: Yes

3. Have the authors made all data underlying the findings in their manuscript fully available?

Reviewer #1: Yes

Reviewer #2: Yes

Reviewer #3: Yes

4. Is the manuscript presented in an intelligible fashion and written in standard English?

Reviewer #1: No

Reviewer #2: Yes

Reviewer #3: Yes

5. Review Comments to the Author

Reviewer #1: The manuscript proposed by Raj, Sifuentes, Kyono and Denver proposes to address an important question of thyroid hormones action in brain during metamorphosis, in the amphibian model Xenopus tropicalis.

By combining RNASeq and ChIPseq data, the authors aim to compare transcriptional variations during natural metamorphosis versus following TH treatment.

Even if the quality of the writing is certain, the manuscript is far from reaching publication quality.

Several subsections of the results section are redundant with other parts of the manuscript. The results description is very concise. Too concise.

For example, the comparison of NF50 vs NF66 stages (lines 291-293) reveals > 4 k DE genes. This represents > 20% of the total number of genes. In fact, this is so high that ANOVA (and more generally parametric models of variance) fail and should not be used. This increases the level of false positives and negatives. If the comparison was indeed NF50 vs NF66, this is a mistake. But if the list of 4k DE genes is the union of two comparisons (untreated vs T3 treatment NF50, and unterated vs T3 treated NF66), this is a different story. This would raise additional questions, but it is at least technologically sound.

This is more than an important technical point, and unfortunately, it is not addressed at all in the manuscript.

- The experimental quality of the data set produced sounds reasonable, but the authors fail to provide additional supporting experimental to help the reader estimate its quality. Not that the work hasn't been done correctly, but ChIP-Seq is notoriously difficult to run, and there are numerous (and unavoidable) caveats at the technical and analysis levels. As the manuscript stands, there are very little data to help the reader estimate how far the data the support the conclusions reached by the authors. This is particularly true for the ChIPSeq, which are always very noisy and where peak height is close to background level.

- Even if this type of comparisons is popular, the comparison of the ChipSeq peaks with the DE genesis highly speculative. The simple fact that a peak is located in a gene (DE or not) is not evidence of direct regulation. The discussion about this point is long, and they rightfully cite the seminal work of Fullwood et al. So why do the authors label this section "identification of direct thyroid hormone target genes using TR Chrip-seq" ? Can we really say this, without additional experimental data (at least 3C or 4C) ?

Minor points

- The tool CLUSTRX looks interesting, and the visual output quite telling. Its source should be referenced.

Reviewer #2: The study by Raj and colleagues is the first genome-wide analysis of gene expression changes for any tadpole tissue spanning the start to end of natural metamorphosis. Theirs is also the first study to directly compare these changes in spontaneous metamorphosis with those induced precociously by TH, a commonly-used approach.

A strong aspect of the current study is that the main conclusions derived from this genome-wide analysis are supported by findings from previous smaller-scale studies. For example, one of the more interesting findings of the study was that, like previous smaller-scale studies, most DEGs in tadpole brain during spontaneous metamorphosis or after T3 treatment did not have TR peaks. The authors put forth several quite plausible explanations for these observations. Another very interesting observation also supported by previous smaller-scale studies was that two thirds of the genes that changed their mRNA levels during spontaneous metamorphosis were not modulated by exogenous T3 in premetamorphic tadpole brain. The authors present several plausible technical and biological explanations for these findings, and conclude that “experiments conducted with exogenous T3 may not be accurate representations of normal developmental processes.” While it is somewhat disappointing that current methods of inducing metamorphosis with T3 and evaluating gene expression do not appear to correspond as well as might be expected with spontaneous metamorphosis, this information is in fact extremely useful to other researchers who use on this method to draw (possibly erroneous) conclusions about natural metamorphosis. One might wonder if inducing metamorphosis instead with a combination of T3 + glucocorticoid may resolve this intriguing discrepancy?

Reviewer #3: In this manuscript Raj et al., have done three complementary experiments on Xenopus metamorphosis: (i) they analysed the gene expression changes in the central nervous system at 4 different time point during spontaneous metamorphosis ; (ii) they perform a classical T3 treatment at 5nM during premetamorphosis and look the gene expression change after 16hr; (iii) last they performed a ChIP-seq experiment to detect the regions of chromatin in which the TRs are binding. These three experiments have been on the central nervous system. The main surprise come from the low number of genes that are found in the three methods: As mentioned by the authors, only half of the genes modulated by treatment of premetamorphic tadpoles with TH changed expression during metamorphosis and these represented only 33% of the genes whose mRNA levels changed during metamorphosis. Similarly, only 24% of genes whose mRNA levels changed during metamorphosis had TR ChIP-seq peaks suggesting that they are direct targets. In the Discussion the authors list all the factors that can explain these low numbers.

This is a very interesting paper, whose conclusions will probably be disliked by many people working on this field, but it is very useful because it clearly reveals the limits of our assays. Despite our fantastic technical abilities, we are still doing very crude and brutal experiments and by doing that we strongly interfere with the systems we are studying.

In addition to this very strong interest, I found the paper well written and the experiments convincing. My main problem is in fact the figures and tables that I find really out of step in terms of quality compared to the text of the paper.

Figure 1B and the relevant text lines 217-228 are a bizarre way of presenting the simple fact that the authors have analyzed 4 stages: I would rather present the developmental serie and would link the various stages to show the comparisons that have been made.

On Figure 2 and 3 I would add heat maps of the 100 or 200 of top genes and would find a way to visualize the genes that are common between spontaneous metamorphosis and TH regulated. Simply put those genes in color in each of the two heat maps.

Figure 4, 5 and 6 are quite boring images of individual genes. I would rather suggest presenting with a scheme the various pathways that are discussed (e.g. cell cycle regulation) and then, again to visualize on each of them the genes that are found by the three technics. There is honestly a strong effort of illustration to do in order to sustain the very nice findings and the great amount of novel information that is present in this paper. Remember that a good part of bioinformatic is to find ways to illustrate very complex and extensive set of information.

Also, there is in my point of view one figure missing. At a minimum we need a Vent diagram to compare the number (and %) of genes that are found associated by each of the three methods. These numbers are discussed lines 294-308 and 400-412 and in the Discussion but we need to see in one clear figure how they relate to each other. What is the number of genes that are regulated/associated in the 3 experiments? Are they TH signalling genes? Could we detect that way new genes that would be very important but have escape attention ?

Minor points

There is an inversion of Figure 5 and 6 with their legends.

Lines 241-244 and Figure 2C: there is not a single gene differentially regulated (positively or negatively) in the three stages? This is curious…

Fig 2D I would be curious to have few examples of the most interesting genes in each of the C1 tp C5 categories. What I have seen in other paper is the line of 3-4 key gene indicated in color with their names. That would be useful.

In our case (in fish) we found many genes implicated in metabolism and a major shift between glycolysis and TCA cycle at different stages during metamorphosis. I think the analysis is a bit short on this point, especially given the importance of the brain as an organ for metabolism also. The link with the appetite control and adipokynes would be super interesting to discuss.

6. PLOS authors have the option to publish the peer review history of their article (what does this mean?). If published, this will include your full peer review and any attached files.

Reviewer #1: No

Reviewer #2: **Yes: **Alexander M Schreiber

Reviewer #3: No

---

## [Author Response · Author response to Decision Letter 0]

6 Jun 2023

Responses to Reviewer Comments

We thank the reviewers for their careful reading of our manuscript and their constructive comments. We have addressed each of the points that were raised as described below. Changes that we made in response to reviewer comments are highlighted in yellow in the revised manuscript file. We also revised the figures as suggested and we rearranged parts of the manuscript to accommodate these changes (we shifted some figure legends and tables).

Reviewer #1

The manuscript proposed by Raj, Sifuentes, Kyono and Denver proposes to address an important question of thyroid hormones action in brain during metamorphosis, in the amphibian model Xenopus tropicalis. By combining RNASeq and ChIPseq data, the authors aim to compare transcriptional variations during natural metamorphosis versus following TH treatment.

Even if the quality of the writing is certain, the manuscript is far from reaching publication quality. Several subsections of the results section are redundant with other parts of the manuscript. 

Question: The results description is very concise. Too concise. For example, the comparison of NF50 vs NF66 stages (lines 291-293) reveals > 4 k DE genes. This represents > 20% of the total number of genes. In fact, this is so high that ANOVA (and more generally parametric models of variance) fail and should not be used. This increases the level of false positives and negatives. If the comparison was indeed NF50 vs NF66, this is a mistake. But if the list of 4k DE genes is the union of two comparisons (untreated vs T3 treatment NF50, and unterated vs T3 treated NF66), this is a different story. This would raise additional questions, but it is at least technologically sound. This is more than an important technical point, and unfortunately, it is not addressed at all in the manuscript.

Answer: Please note that we used one-way ANOVA only to analyze the targeted RTqPCR data as we indicated in the Materials and Methods. As described in our original publications (Kyono et al., 2020; Wen et al., 2019), we conducted differential expression analysis using the R package DESeq2 (v1.22.0) to identify differentially expressed genes. We reanalyzed the data for the current manuscript using the most recent Xenopus tropicalis genome build (v.9.1). The list of 4K DEGs is not a union of two comparisons as the reviewer indicates; that is, it did not involve the T3 treated vs untreated (NF50) dataset, only the developmental series (NF50, NF56, NF62 and NF66). And there was no T3 treatment of NF66 animals, only of NF50. The NF66 stage is the completion of metamorphosis, and T3 treatment has no developmental/morphological effects in the newly metamorphosed frogs.   

Question: The experimental quality of the data set produced sounds reasonable, but the authors fail to provide additional supporting experimental to help the reader estimate its quality. Not that the work hasn't been done correctly, but ChIP-Seq is notoriously difficult to run, and there are numerous (and unavoidable) caveats at the technical and analysis levels. As the manuscript stands, there are very little data to help the reader estimate how far the data the support the conclusions reached by the authors. This is particularly true for the ChIPSeq, which are always very noisy and where peak height is close to background level.

Answer: We agree that ChIP-seq (and related techniques) are technically challenging. We have a track record for conducting such studies using antibodies (ChIP-seq), chromatin streptavidin precipitation (ChSPseq) and methyl capture sequencing (MethylCap-seq) (Avila-Mendoza et al., 2020; Knoedler et al., 2017; Kyono et al., 2020). We indicated in the manuscript (original lines 205-206, revised lines 195-201) that we provided targeted ChIP-qPCR validations for the TR ChIP-seq experiment in a previous manuscript published in Endocrinology (Raj et al., 2020) (see both the main Endocrinology manuscript and the supporting information for the manuscript). These included six TH-TR target genes (thrb, thibz, klf9, gadd45�, tet2, tet3) and two negative control regions (ifabp promoter, thrb exon 5). We now include validations for three additional genomic regions in the supporting information of the current manuscript (Supplemental Fig. 2). Lastly, we used a high stringency FDR cutoff (0.01) for our TR ChIP-seq analysis which reduced chances for false positives. 

Question: Even if this type of comparisons is popular, the comparison of the ChipSeq peaks with the DE genesis highly speculative. The simple fact that a peak is located in a gene (DE or not) is not evidence of direct regulation. The discussion about this point is long, and they rightfully cite the seminal work of Fullwood et al. So why do the authors label this section "identification of direct thyroid hormone target genes using TR Chrip-seq" ? Can we really say this, without additional experimental data (at least 3C or 4C) ?

Answer: We agree with the reviewer. We think that we are forthright in describing how and why we selected the criteria that we did to assign TR peaks to genes, and we discussed the caveats for assigning putative TREs (TR peaks) to genes (as the reviewer mentioned). We note that all previously validated TREs in Xenopus have been found within genes, within 5’ UTRs or in the 5’ flanking region, which is why we selected the genomic range (10 kb upstream of the TSS and within gene bodies) that we did to provisionally assign peaks to genes. We agree with the reviewer that more direct evidence is needed to assign functional TREs to genes. We are cognizant of such issues, and earlier we used ChIA-PET data that we validated by 3C assay to show that an upstream TRE in the Xenopus Klf9 gene interacts with the transcription start site (Bagamasbad et al., 2015), and we were the first to use CRISPR-Cas9 genome editing to mutate a putative TRE to demonstrate its transactivation function (in the mouse Dnmt3a gene) (Kyono et al., 2016). We agree that this kind of evidence is required to formally assign a regulatory element to a gene, and that it is premature to conclude that genes with proximate TR peaks are direct thyroid hormone target genes. Therefore, we have changed this label and modified some of the text. 

Minor points

Question: The tool CLUSTRX looks interesting, and the visual output quite telling. Its source should be referenced.

Answer: The software is called Clust (not CLUSTRX; this was a mistake, thank you). We have now included the reference in the manuscript.

Reviewer #2

The study by Raj and colleagues is the first genome-wide analysis of gene expression changes for any tadpole tissue spanning the start to end of natural metamorphosis. Theirs is also the first study to directly compare these changes in spontaneous metamorphosis with those induced precociously by TH, a commonly-used approach.

A strong aspect of the current study is that the main conclusions derived from this genome-wide analysis are supported by findings from previous smaller-scale studies. For example, one of the more interesting findings of the study was that, like previous smaller-scale studies, most DEGs in tadpole brain during spontaneous metamorphosis or after T3 treatment did not have TR peaks. The authors put forth several quite plausible explanations for these observations. Another very interesting observation also supported by previous smaller-scale studies was that two thirds of the genes that changed their mRNA levels during spontaneous metamorphosis were not modulated by exogenous T3 in premetamorphic tadpole brain. The authors present several plausible technical and biological explanations for these findings, and conclude that “experiments conducted with exogenous T3 may not be accurate representations of normal developmental processes.” While it is somewhat disappointing that current methods of inducing metamorphosis with T3 and evaluating gene expression do not appear to correspond as well as might be expected with spontaneous metamorphosis, this information is in fact extremely useful to other researchers who use on this method to draw (possibly erroneous) conclusions about natural metamorphosis. One might wonder if inducing metamorphosis instead with a combination of T3 + glucocorticoid may resolve this intriguing discrepancy?

Answer: We thank the reviewer for their supportive comments. It might be interesting to analyze the combined T3 + glucocorticoid treatment. However, we believe that the possible artifactual responses are more likely related to the pharmacological effect of exposing premetamorphic tadpoles to a metamorphic climax stage dose of T3 (or even greater, as we found previously that tadpoles concentrate hormone from their environment, and we discuss in the text). A dose response experiment with T3 (and a time course) might resolve this issue. 

Reviewer #3

In this manuscript Raj et al., have done three complementary experiments on Xenopus metamorphosis: (i) they analysed the gene expression changes in the central nervous system at 4 different time point during spontaneous metamorphosis ; (ii) they perform a classical T3 treatment at 5nM during premetamorphosis and look the gene expression change after 16hr; (iii) last they performed a ChIP-seq experiment to detect the regions of chromatin in which the TRs are binding. These three experiments have been on the central nervous system. The main surprise come from the low number of genes that are found in the three methods: As mentioned by the authors, only half of the genes modulated by treatment of premetamorphic tadpoles with TH changed expression during metamorphosis and these represented only 33% of the genes whose mRNA levels changed during metamorphosis. Similarly, only 24% of genes whose mRNA levels changed during metamorphosis had TR ChIP-seq peaks suggesting that they are direct targets. In the Discussion the authors list all the factors that can explain these low numbers.

This is a very interesting paper, whose conclusions will probably be disliked by many people working on this field, but it is very useful because it clearly reveals the limits of our assays. Despite our fantastic technical abilities, we are still doing very crude and brutal experiments and by doing that we strongly interfere with the systems we are studying.

In addition to this very strong interest, I found the paper well written and the experiments convincing. My main problem is in fact the figures and tables that I find really out of step in terms of quality compared to the text of the paper.

Answer: We thank the reviewer for their supportive comments. We have now revised the figures and tables as described below.

Question: Figure 1B and the relevant text lines 217-228 are a bizarre way of presenting the simple fact that the authors have analyzed 4 stages: I would rather present the developmental series and would link the various stages to show the comparisons that have been made.

Answer: The purpose of the original figure 1B was to show the comparisons that we made among the RNA-seq data from the 4 developmental stages, but we agree that this attempt was not the best. To aid readers who may not be familiar with the gross morphological changes that occur during metamorphosis of X. tropicalis we now include tadpole images on the new figure 1B. The text on lines 217-228 (now lines 213-218) simply states the comparisons that we made between the mRNA levels determined by RNA-seq at the 4 developmental stages. 

Question: On Figure 2 and 3 I would add heat maps of the 100 or 200 of top genes and would find a way to visualize the genes that are common between spontaneous metamorphosis and TH regulated. Simply put those genes in color in each of the two heat maps.

Answer: We now provide heatmaps that show the top 100 upregulated and the top 100 downregulated genes during metamorphosis in figure 2. Although these heatmaps do not provide additional information beyond what we presented in the clustering analysis, which includes many more genes (Fig. 1), they may help the reader visualize the gene expression changes that occur during metamorphosis of tadpole brain. However, we cannot generate a heat map comparing expression levels between the developmental series and the +/- T3 treatment since these were two separate experiments.

Question: Figure 4, 5 and 6 are quite boring images of individual genes. I would rather suggest presenting with a scheme the various pathways that are discussed (e.g. cell cycle regulation) and then, again to visualize on each of them the genes that are found by the three technics. There is honestly a strong effort of illustration to do in order to sustain the very nice findings and the great amount of novel information that is present in this paper. Remember that a good part of bioinformatic is to find ways to illustrate very complex and extensive set of information.

Answer: We now include a KEGG pathway analysis diagram for cell cycle in figure 4, and for ribosome biogenesis in figure 5. We chose to not provide similar pathway figures for the T3 experiment because we do not trust that such analyses are reliable representations of normal developmental processes, as we discuss in the manuscript. 

We do not provide GO or pathway analyses for the TR ChIP-seq experiment because we cannot determine if genes with proximate TR peaks are actually regulated by the T3-TR complex. Indeed, our analysis showed that 2/3 of the TR peaks associated with genes were found proximate to genes that did not change expression during metamorphosis (∼5/6 of TR peaks were not associated with genes regulated by exogenous T3). 

Therefore, the only dataset that we think is amenable to such analysis is the developmental series, which includes differentially expressed genes that are likely to be involved in the biological processes under investigation. 

We decided to keep figures 6 and 7 since they depict changes in the expression of key genes involved with neural differentiation and neuroendocrine function, which we think that researchers working on Xenopus will find interesting. 

Question: Also, there is in my point of view one figure missing. At a minimum we need a Vent diagram to compare the number (and %) of genes that are found associated by each of the three methods. These numbers are discussed lines 294-308 and 400-412 and in the Discussion but we need to see in one clear figure how they relate to each other. What is the number of genes that are regulated/associated in the 3 experiments? Are they TH signalling genes? Could we detect that way new genes that would be very important but have escape attention?

Answer: We now provide additional an Venn diagram in figure 2 comparing differentially expressed genes during spontaneous metamorphosis and after T3 treatment of premetamorphic tadpoles. However, we cannot generate Venn diagrams that also include the TR ChIP-seq data because the measurements are different (i.e., DEGs vs. TR peaks). Table 5 summarizes the TR ChIP-seq data. Most of the TR peaks were in proximity to genes that did not change during metamorphosis or after T3 treatment. These genes may not be regulated by T3-TR at all, or they may not be regulated by T3-TR during the developmental stages that we analyzed; or, the TREs at which the TRs associate regulate genes other than those with which they are in close proximity (see Discussion and our answer to reviewer #1 above).

There are several genes involved with T3 action that we highlight in the manuscript. It is possible that other genes in our lists play a role in T3 signaling, but we have no way of determining this from our analyses. We provide extensive supplemental tables that we hope will be a valuable resource for future investigations. 

Minor points

Question: There is an inversion of Figure 5 and 6 with their legends.

Answer: Thank you for noticing this error which has now been corrected.

Question: Lines 241-244 and Figure 2C: there is not a single gene differentially regulated (positively or negatively) in the three stages? This is curious…

Answer: Thank you for noticing this omission which has now been corrected. There were 49 genes that were common between the three developmental intervals (NF50-56, NF56-62 and NF62-66). This number is low in part because the number of genes regulated between NF62-66 was low (364). 

Question: Fig 2D I would be curious to have few examples of the most interesting genes in each of the C1 tp C5 categories. What I have seen in other paper is the line of 3-4 key gene indicated in color with their names. That would be useful.

In our case (in fish) we found many genes implicated in metabolism and a major shift between glycolysis and TCA cycle at different stages during metamorphosis. I think the analysis is a bit short on this point, especially given the importance of the brain as an organ for metabolism also. The link with the appetite control and adipokynes would be super interesting to discuss.

Answer: The expression patterns of some of these genes are shown in figures 5-7. We chose these genes to plot because of their previously demonstrated (or hypothesized based on studies in other species) importance for Xenopus development. We think that Xenopus researchers will be interested in the developmental profiles of these genes, which we categorized based on their functions in discrete developmental/cellular signaling pathways. We agree that genes involved with energy balance and the regulation of food intake are interesting, which we present briefly. However, one could make a similar case for reproduction, pituitary regulation, among others. We think that these are interesting discussion points, but to do them justice would require that the manuscript be considerably longer than it already is, and so we think that they are beyond the scope of the current manuscript. 

References Cited

Avila-Mendoza, J., Subramani, A., Sifuentes, C.J., Denver, R.J., 2020. Molecular Mechanisms for Kruppel-Like Factor 13 Actions in Hippocampal Neurons. Molecular Neurobiology 57, 3785-3802.

Bagamasbad, P., Bonett, R., Sachs, L., Buisine, N., Raj, S., Knoedler, J., Kyono, Y., Ruan, Y., Ruan, X., Denver, R., 2015. Deciphering the regulatory logic of an ancient, ultraconserved nuclear receptor enhancer module. Molecular Endocrinology 29, 856-872.

Knoedler, J.R., Subramani, A., Denver, R.J., 2017. The Kruppel-like factor 9 cistrome in mouse hippocampal neurons reveals predominant transcriptional repression via proximal promoter binding. BMC Genomics 18, 299.

Kyono, Y., Raj, S., Sifuentes, C.J., Buisine, N., Sachs, L., Denver, R.J., 2020. DNA methylation dynamics underlie metamorphic gene regulation programs in Xenopus tadpole brain. Dev. Biol. 462, 180-196.

Kyono, Y., Subramani, A., Ramadoss, P., Hollenberg, A.N., Bonett, R.M., Denver, R.J., 2016. Liganded thyroid hormone receptors transactivate the DNA methyltransferase 3a gene in mouse neuronal cells. Endocrinology 157, 3647-3657.

Raj, S., Kyono, Y., Sifuentes, C.J., Arellanes-Licea, E., Subramani, A., Denver, R.J., 2020. Thyroid hormone induces DNA demethylation in Xenopus tadpole brain. Dryad Digital Repository Deposited 1 July, 2020.

Wen, L., He, C., Sifuentes, C.J., Denver, R.J., 2019. Thyroid hormone receptor alpha Is required for thyroid hormone-dependent neural cell proliferation during tadpole metamorphosis. Frontiers in Endocrinology 10.

---

## [Editor Report · Decision Letter 1]

14 Jun 2023

Metamorphic gene regulation programs in Xenopus tropicalis tadpole brain

PONE-D-23-06337R1

Dear Dr. Denver,

We’re pleased to inform you that your manuscript has been judged scientifically suitable for publication and will be formally accepted for publication once it meets all outstanding technical requirements.

Kind regards,

Laurent Coen, Ph.D.

Academic Editor

PLOS ONE

Additional Editor Comments (optional):

After a careful reading of the corrections made by the authors, I consider that all the points raised by the reviewers have been taken into account.

All the modifications and comments asked by the reviewers have been addressed in the revised version of the manuscript, and important changes in the text and in figs presentation have been done.

Responses to comments, rewordings of the text and changes to the figures have clarified the paper's weaknesses pointed out in its initial version, and it can therefore be accepted for publication in its revised form, without the need for further consideration by the reviewers.
---

## [Editor Report · Acceptance letter]

19 Jun 2023

PONE-D-23-06337R1 

Metamorphic gene regulation programs in *Xenopus tropicalis* tadpole brain 

Dear Dr. Denver:

I'm pleased to inform you that your manuscript has been deemed suitable for publication in PLOS ONE. Congratulations! Your manuscript is now with our production department. 

Kind regards, 

on behalf of

Dr. Laurent Coen 

Academic Editor

PLOS ONE